# Efficient Reasoning Through Suppression of Self-Affirmation Reflections in Large Reasoning Models

## Abstract

While recent advances in large reasoning models have demonstrated remarkable performance, efficient reasoning remains critical due to the rapid growth of output length. Existing optimization approaches highlights a tendency toward "overthinking", yet lack fine-grained analysis. In this work, we focus on **Self-Affirmation Reflections**: redundant reflective steps that affirm prior content and often occurs after the already correct reasoning steps. Observations of both original and optimized reasoning models reveal pervasive self-affirmation reflections. Notably, these reflections sometimes lead to longer outputs in optimized models than their original counterparts. Through detailed analysis, we uncover an intriguing pattern: compared to other reflections, the leading words (i.e., the first word of sentences) in self-affirmation reflections exhibit a distinct probability bias. Motivated by this insight, we can locate self-affirmation reflections and conduct a train-free experiment demonstrating that suppressing self-affirmation reflections reduces output length without degrading accuracy across multiple models (R1-Distill-Models, QwQ-32B, and Qwen3-32B). Furthermore, we also improve current train-based method by explicitly suppressing such reflections. In our experiments, we achieve length compression of 18.7% in train-free settings and 50.2% in train-based settings for R1-Distill-Qwen-1.5B. Moreover, our improvements are simple yet practical and can be directly applied to existing inference frameworks, such as vLLM. We believe that our findings will provide community insights for achieving more precise length compression and step-level efficient reasoning.

## 1 Introduction

Recent advancements [23, 43, 2, 17] in large language models have delivered remarkable performance across various tasks [10, 4], yet they still struggle with complex reasoning tasks demanding advanced mathematical [15, 22] and formal logical deduction [35] abilities. Works [18, 17, 12, 38, 36] like Deepseek-R1 [12] have driven progress in reasoning models via reinforcement learning. However, current large reasoning models are troubled by high token usage and computational costs due to generating lengthy reasoning chains. Researchers have been delving into different technical strategies to enhance reasoning efficiency, like refining latent representations [14, 33, 7] and combining small models [27, 11]. In order to directly optimize the model itself, several approaches have emerged as promising methods, using supervised fine-tuning (SFT) [41, 13, 25], direct preference optimization (DPO) [32], and reinforcement learning (RL) [3, 24, 3, 1, 44].

Although these methods have yielded impressive results, such as producing more semantically efficient or direct solutions, we observed that redundant reflection persists in the optimized models [3, 24]. Rather than attributing this phenomenon to the more general "overthink" issue (defined in

work [34] as the generation of excessively detailed or unnecessary lengthy reasoning steps by large reasoning models during inference, which undermines reasoning efficiency), we conducted a more detailed examination of these examples (for more details, refer to *Section 3*). As illustrated in Figure 1, we have identified a common phenomenon: affirming prior content typically occurs following correct steps, which we refer to as Self-Affirmation Reflection. In some cases, this superfluous reflection even leads to longer answers for certain problems in the optimized models.

Since this phenomenon is also widespread in the optimized models [3, 24], such reflection essentially originates from the base models. Therefore, we investigate scenarios where original reasoning models frequently execute self-affirmation reflection. Empirical analysis reveals a notable phenomenon: During repeated rollout processes at reflection points, models do not consistently engage in reflection behavior, as illustrated in the yellow box of Figure 1. In fact, after generating correct intermediate steps, models often exhibit ambiguity in transitioning between problem-solving steps and reflection initiation. Since prior works [31, 26] have established that certain keywords, termed leading words in this paper, significantly influence a model's decision to trigger the reflection mechanism. To characterize the self-affirmation reflection type, we conduct statistical analysis comparing its leading word distribution to those of other reflection categories. As shown in *Section 4*, self-affirmation reflection exhibits statistically significant divergence in leading word distribution patterns.

> How many ways can we put 3 math books and 5 English books on a shelf if all the math books must stay together and all the English books must also stay together? (The math books are all different and so are the English books.)

> Okay, so I have this problem here: I need to figure out how many ways I can arrange 3 math books and 5 English books on a shelf. The catch is that all the math books have to stay together, and ⋯
> ⋯
> Wait, let me double-check my reasoning. So, treating the math books as one block and the English books as another block, we have 2 blocks. The number of...
> ⋯
> Wait, but in the first approach, the math block can be placed in 6 different starting positions, and the English block is placed in the⋯
> So, $6 \times 3! \times 5! = 6 \times 6 \times 120 = 4320$. Yes, that's correct.
> ⋯

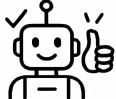
> [Rollout 1] Wait, let me double-check⋯
> [Rollout 2] Let me double-check⋯
> [Rollout 3] But wait, let me double-check⋯.
> [Rollout 4] Just to make sure I didn't make a mistake, let me think through it again....

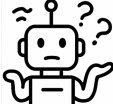
> [Rollout 1] Wait, but in the first approach, the ⋯
> [Rollout 2] So, $6 \times 6 \times 120 = 4320$.
> [Rollout 3] Therefore, $6 \times 3! \times 5! = 6 \times 6 \times 120 = 4320$.
> [Rollout 4] Thus, $6 \times 3! \times 5! = 6 \times 6 \times 120 = 4320$.

Figure 1: **Self-Affirmation Reflection** phenomenon. We investigate the characteristics when the reflection occurs by performing multiple rollouts at reflection points (marked by the colored "Wait"). As illustrated in the figure below, the model sometimes consistently performs a reflection (marked in pink box), while at other times it takes different actions (marked in yellow box). Since the latter reflections generally serve to reaffirm previous content, we refer to this behavior as the self-affirmation reflection.

Building on this insight, we identify conditions under which self-affirmation reflections occur during model reasoning and present a straightforward yet effective method that explicitly suppresses rethinking confidence at these crucial confusion points. Notably, our approach entails no extra cost and is extremely simple to implement. It can be seamlessly integrated into current inference frameworks such as vLLM [19]. Under a train-free experimental setup, while maintaining nearly the same performance level, we have achieved length reductions of 18.7%, 14.3%, 11.1%, 9.1%, and 8.4% for R1-Distill-Qwen-1.5B/7B/32B, QwQ-32B, and Qwen3-32B respectively, and in some cases, even better performance is obtained. Furthermore, we apply our method to a representative train-based approach [3], which generates responses of varying lengths by model self-sampling. Correct responses are prioritized, with shorter lengths receiving higher rewards. Our method complements this strategy by explicitly suppressing self-affirming reflections within positive samples and achieve significantly shorter outputs with competitive performance. Extensive experiments in *Section 5* across different settings validate the feasibility and significance of the findings in this paper.

Our work offers three key benefits: (1) We provides the first focused analysis of the Self-Affirmation Reflection phenomenon and **provides actionable insights for improving reasoning models**. (2) **We present a simple yet practical intervention strategy**: suppressing crucial leading tokens. This directly lessens Self-Affirmation Reflection, achieving efficient output compression without modifying the model architecture or training process. (3) Extensive experiments across diverse models and datasets in **both train-free and train-based settings** demonstrate that mitigating Self-Affirmation Reflection can effectively reduce output length while preserving or even enhancing performance.

## 2 Related Work

Recent surveys [29, 39, 34] have summarized progress in efficient reasoning. In this section, we highlight representative methods to provide a concise overview of key developments in this area.

The first category of methods [13, 42, 20] aim to obtain some information from the input to estimate the cost in advance. Some methods [13, 42, 20] choose to estimate the reasoning cost. For example, TokenBudget [13] instructs models to estimate the token count required to solve a problem upfront, compelling them to generate concise responses within a predefined budget. Other methods [27, 5, 9, 8] estimate the model size. Approaches like RouterLLM [27] and Sot [5] pre-train classifiers to route problems to the most suitable LLM based on task complexity. In contrast, router-free methods such as Self-ReF [9] and Seek-Router [8] predict the necessity of invoking additional LLMs by analyzing internal uncertainty scores. However, how to accurately determine the optimal cost is still a problem.

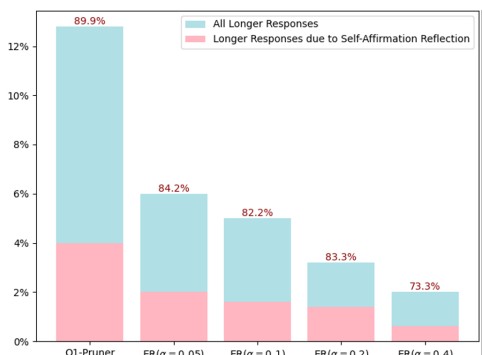

Figure 2: The ratio of questions with longer average response lengths in MATH500 [22] when testing released checkpoints from EfficientReasoning [3] and O1-Pruner [24](marked as blue bar). Through manual inspection, we additionally show the proportion of responses that are longer due to output self-affirmation reflections (marked as pink bar). Moreover, the accuracy on MATH500 is labeled at the top of each bar.

Recent advances in the second category of methods have focused on refining the output of optimization models. As demonstrated in [28], simply adding intermediate "thinking" tokens, or even nonsensical fillers like "......", can lead to satisfactory performance. Subsequent works, such as those in [14, 33, 7], have explored treating the last layer hidden states of LLMs as "continuous thinking" to replace traditional discrete markers. These approaches optimize latent representations through training. However, such methods may render parts of the thought chain invisible, potentially introducing security risks. In addition to optimizing the output sequence, there are also methods to choose to switch the model during output, such as RSD [21] integrating multiple LLMs to achieve dynamic reasoning. These methods demand precise determination of when to switch models and which model to switch to.

The third category of methods centers on fine-tuning LLM itself to directly address redundancy in the output of the original model. Generally, these methods [24, 3, 3, 1, 44] first obtain responses of varying lengths via model self-sampling. Then, they design a length-based reward conditioned on correctness, with shorter correct answers receiving higher rewards. However, balancing accuracy and output length remains a significant challenge. Although these methods reduce average response length of the entire datasets, they do not guarantee shorter outputs across all instances. Specifically, the figure 2 highlights that at the instance level, some outputs (depicted in blue) remain longer than those from the original model. Our train-based experiments in Section 5.2 is closely related to these methods but differs in focus. Rather than focusing on reward function design, our objective is to directly optimize sampling outcomes, thereby refining the quality of the constructed data pairs and achieving more succinct responses.

## 3 Observation

Our observation begins with experiments evaluating two representative open-source works [24, 3]. We tested R1-Distill-Qwen-1.5B [12] and QwQ-Preview [37] on the MATH500 dataset [22] alongside their compressed counterparts (EfficientReasoning [3] and O1-Pruner [24]). For each question, three responses were sampled, with average response length recorded.

As shown in Figure 2, results exhibit a counterintuitive trend: certain ratio of instance-level questions elicit longer responses, yet most of these extended responses remain correct (e.g., 91.6% in QwQ-Preview settings). This pattern persists across varying compression levels. For instance, the parameter

146 $\alpha$ in EfficientReasoning[3] balances brevity and accuracy. Larger $\alpha$ values shorten outputs at the cost
147 of reduced accuracy. As shown in Figure 2, even at mild compression ($\alpha = 0.05$), 6% of questions
148 generated longer responses. Aggressive compression ($\alpha = 0.4$) still failed to resolve this behavior,
149 though the trade-off led to a significant drop in accuracy.

150 To gain deeper insights into the optimized models, we conducted a manual examination and analyzed
151 their qualitative behaviors, as shown in Figure 3. Two key characteristics consistently emerge: (1) the
152 model tends to produce more concise steps during the problem-solving phase. For instance, the brown
153 text in Figure 3 illustrates how the optimized model succinctly summarizes the problem background.
154 (2) certain questions have been solved using more direct approaches. This arises because, during the
155 self-sampling process, the model generates multiple potential problem-solving methods. By favoring
156 shorter solution ideas, the model discards superfluous strategies.

157 However, we also observe that the model frequently engages in prolonged reflections even after
158 executing correct reasoning steps or arriving at the correct answer, resulting in excessively lengthy
159 outputs. Figure 3 illustrates this issue: compared to the baseline R1-Distill-Qwen-1.5B model [12],
160 the optimized EfficientReasoning model ($\alpha = 0.05$) [3] achieves the correct solution more rapidly
161 but produces lengthier responses due to increased self-reflections. These reflections often show a
162 tendency to affirm the previous content. Quantitative results in Figure 2 further confirm that this
163 pattern is prevalent across current efficient reasoning models [3, 24]. Overall, these findings suggest
164 that naively optimizing for solution-level simplicity is not enough to effectively compress the model's
165 outputs. This indicates that existing frameworks may still have opportunities to enhance output
166 simplicity by suppressing step-level self-affirmation reflections.

167 Notably, while this phenomenon was widely summarized as an overthink problem in previous works
168 [29, 39, 34], its persistence in optimized reasoning models underscores the need for targeted solutions.
169 Therefore, we formally define this behavior as self-affirmation reflection, as introduced in Section
170 1, and focus on it in this paper. To address this issue fundamentally, we analyze how to locate and
171 suppress self-affirmation reflection in the original model, as detailed in the subsequent Section 4.

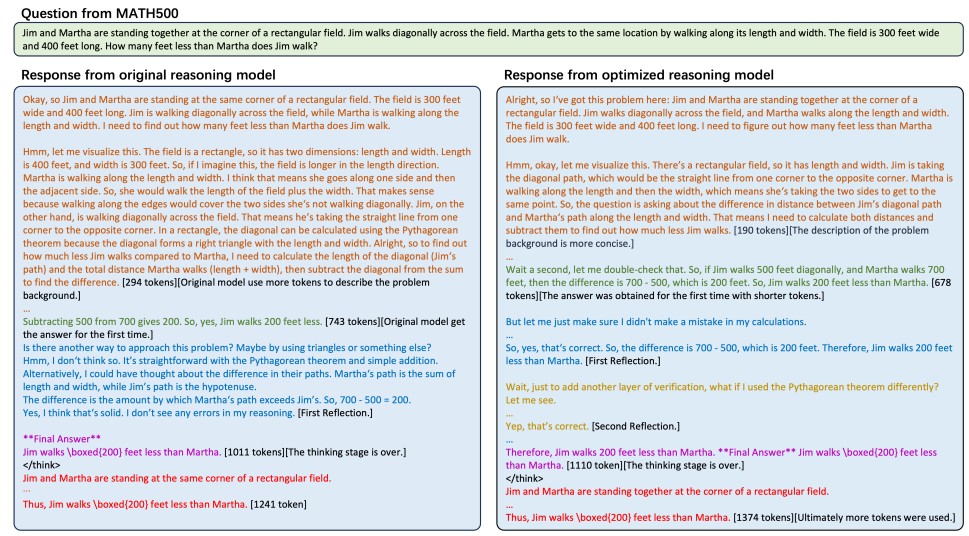

Figure 3: Qualitative analysis of responses from original and optimized R1-Distill-Qwen-1.5B. The
colored text represents the original response contents of the two models, and the black text indicates
our description of the progress of solving the problem. Provide the number of tokens that have been
used when necessary.

## 4 Analysis

### 4.1 Locate Self-Affirmation Reflections

174 To investigate the self-affirmation reflection, we first randomly sampled 500 training instances from
175 the MATH dataset. For each problem, we generated a single solution using R1-Distill-1.5B [12]. The
176 generated reasoning steps were split using the "$\backslash n \backslash n$" delimiter.

Then, we utilized the Qwen2.5-72B-Instruct [43] to classify each step as either a reasoning reflection or non-reflective reasoning. To improve the accuracy, we further combine the previous method [6] to assist in judgment. For steps identified as reasoning reflections, we invoked the Qwen2.5-72B-Instruct again to determine whether the reflection affirmed the preceding reasoning, i.e., whether it constituted a self-affirmation reflection. The specific prompts used for classification are detailed in Appendix A.3. To validate this automated classification, we manually annotated self-affirmation reflections in all responses of 20 problems to form a test set. Evaluation revealed the Qwen2.5-72B-Instruct achieved an accuracy of 80.6%. This is primarily because while most self-affirmation reflections contain explicit signal words such as "that's correct" in Figure 3, some self-affirmation reflections merely repeat previous conclusions to indicate affirmation of prior content, making these more challenging to detect.

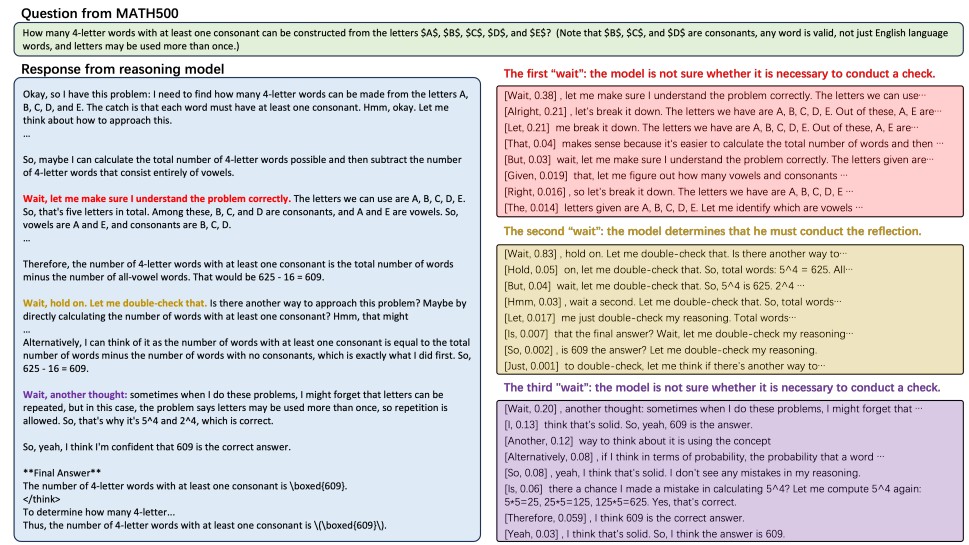

Figure 4: A qualitative analysis result of self-affirmation reflections and other reflections. At the beginning of each reflective sentence, we save the top 8 high-probability words and continue rollout based on these words. We also marked the first word and the corresponding confidence level. For more details, please refer to Section 4.1.

Based on the results of classification, our analysis proceeded from two complementary perspectives to identify patterns in self-affirmation reflection occurrence. **First**, prior researches [31, 26] has established that language patterns are critical in determining whether models activate specific reflection mechanisms during decision-making. Therefore, we sampled 20 self-affirmation reflections and other reflection types, then performed multiple rollout iterations from their contextual positions. We hope to explore whether models can elicit distinct behavioral responses by producing diverse leading words. As shown in Figure 4, we found that self-affirmation reflections exhibit significantly greater diversity in behavior-guiding continuation patterns compared to other reflection types.

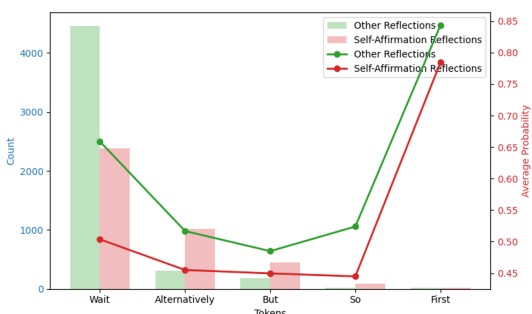

Figure 5: The frequency (via bar chart) and average confidence scores (via line plot) of the first words in all reflective sentences. The top five words are presented in our results. We found that when the model engages in self-affirmation reflections, it exhibits a certain level of uncertainty compared to other reflections.

Specifically, during the initial reflection (Self-Affirmation Reflection), the model attempts to verify its understanding of the problem context. Examining the top 8 predicted word continuations, we observe the model's indecision between further reflection and progressing toward a solution, as evidenced by suggestions such as "break it down". In the second reflection (Other Reflection), the trajectories of the model show a high degree of consistency. Here, the top 8 predicted words uniformly advocate for continued reflection, such as "double-check". The final reflection (Self-Affirmation Reflection) occurs after the model has

already provided the correct answer. Similar to the first reflection, we observe the model confronting uncertainty, contemplating whether to terminate the task or initiate another round of reflection.

**Second**, we conducted an additional analysis of the quantity and probability of the leading words across all reflections. It is worth noting that, for simplicity, we only selected the first word as leading words, but a typical leading word is sometimes not necessarily a single word. For example, when sentences start with "But", we found that the next word is "wait" in 38% of self-affirmation reflections. In this case, "But wait" is more suitable to be used as a unique leading word. During statistical processing, we excluded repetitive tail-end reflections from model outputs. Because self-affirmation reflections sometimes exhibited recurring patterns where incremental reinforcement of leading-word probabilities led to closed-loop generation. These loops could skew statistical accuracy. While this phenomenon warrants further study, it falls outside the scope of this work. For an illustrative example of such looping behavior and comprehensive results including looped sentences, refer to Appendix A.4.

The final result is shown in Figure 5. We present the top 5 high-frequency words in self-affirmation reflection. To our surprise, the confidence level of leading words in self-affirmation reflection appears to be lower than that in other Reflections. This finding echoes the results from the Figure 4, indicating that when the model is uncertain about its actions, the output leading words also lack confidence.

### 4.2 Suppress Self-Affirmation Reflections

Up to now, our analysis highlights a critical insight: leading tokens in self-affirmation reflections exhibit significantly lower generation probabilities compared to other reflection types. Our objective is to suppress self-affirmation reflections in order to achieve shorter outputs while maintaining performance. Consequently, we propose a straightforward intervention: when the model assigns a low probability to generating leading words, we set their probability to zero to suppress self-affirmation reflections. Given the statistical significance of the "Wait" token, we first focus on it in this paper.

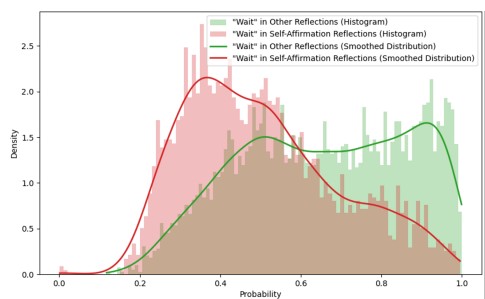

Figure 6: The distribution of "Wait" in the two types of reflections.

To validate our method, we analyze the probability density of the "Wait" token across two reflection types in Figure 6. Though their distributions exhibit partial overlap, they remain distinguishable through thresholding. We acknowledge that suppressing low-probability "Wait" instances may inadvertently filter out other reflections. However, empirical results show that even when such instances are suppressed, other high-probability tokens can still facilitate emergence of necessary reflections via compensatory mechanisms, as exemplified by the second reflection in Figure 4. We also recognize the limitation of exclusively targeting the "Wait" token, which relates to balancing interventions across multiple leading words. In practical results, we found that since "wait" frequently follows "But", suppressing this token also partially mitigates self-affirmation reflections. Consequently, we intervene on both "Wait" and "wait" (collectively referred to as the "wait" tokens hereafter). **More discussions of other interfered tokens appear in Appendix A.5.**

Our results also demonstrate that suppressing low-probability "wait" tokens maintains performance integrity and produces shorter outputs. In train-free settings (e.g., R1-Distill-Qwen-1.5B in Table 1), output lengths shorten while performance improves when thresholds decrease below 0.5. For training-based scenarios, ablation studies identify optimal performance at a threshold of 0.3. These findings curiously align with the probability distributions in Figure 6, suggesting that thresholds $\leq 0.3$ effectively suppress self-affirmation reflections while having minimal impact on other reflection types.

## 5 Experiments

We first assess the impact of removing Self-Affirmation Reflections in Section 5.1. Subsequently, in Section 5.2, we integrate a representative train-based approach to verify whether explicitly suppressing self-affirmation reflections can lead to improved results.

### 5.1 Train-free Experiments

#### 5.1.1 Settings

We hypothesize that removing low-probability self-affirmation reflection will not degrade model performance. To test this hypothesis, we conducted experiments across several prominent reasoning models: R1-Distill-Qwen-1.5B/7B/32B [12], QwQ-32B [37] and Qwen3-32B [38]. We also additionally compared Underthink [40], which intervened with logits within a certain window length. For more details and our discussion on this work, please refer to Appendix A.2. Similar to the previous approaches [6, 3, 24, 32], our experiments are conducted on four datasets: (1) MATH500 [22], comprising 500 problems across seven mathematical domains (algebra, geometry, number theory, etc.) that challenge both humans and LLMs; (2) AIME24, featuring 2024 American Invitational Mathematics Examination problems to test advanced mathematical competition problem-solving skills; (3) AMC23 [15], with problems from the 2023 American Mathematics Competitions covering secondary school mathematics; and (4) GSM8K [10], containing 8.5K linguistically diverse elementary school math problems to evaluate simple arithmetic reasoning. **Regarding the results of the out-of-domain dataset, please refer to the Appendix A.6.**

For AIME24 and AMC23, we sampled eight responses per question. For MATH500 and GSM8K, we collected one response per problem. The maximum allowed output length was 32k tokens. In order to evaluate Qwen3-32B [38], we utilized the latest vLLM implementation [19] in zero-shot inference settings. We report the average accuracy (Acc) and average token count (LEN) per response across all datasets. All experiments were conducted on NVIDIA L20 GPUs.

#### 5.1.2 Results

As shown in Table 1, we evaluated the impact of different thresholds on performance. Our results indicate that an appropriate threshold can effectively reduce output length without compromising performance, and in some cases, even enhances performance with shorter outputs, which is consistent with our previous analysis in Figure 6. However, as the threshold increases, high-probability "wait" tokens are also filtered out, leading to a noticeable decline in performance due to the removal of some necessary reflections. Despite its simple implementation, our method achieves a significant reduction in output length with almost no performance loss. Specifically, it reduces average length by 18.7%, 14.3%, 11.1%, 9.1% and 8.4% for R1-Distill-Qwen-1.5B (Threshold-0.9), R1-Distill-Qwen-7B (Threshold-0.7), R1-Distill-Qwen-32B (Threshold-0.7), QwQ-32B (Threshold 0.7) and Qwen3-32B (Threshold 0.9), respectively. Moreover, we were surprised to find that the compression ratio adapts dynamically across different datasets. For instance, AIME24 demonstrates a lower compression ratio than MATH500. This aligns with the intuition that AIME24, being more challenging, inherently possesses less compressible structure compared to MATH500.

### 5.2 Train-based Experiments

#### 5.2.1 Settings

In this section, we incorporate our approach into a representative training-based method. We utilize EfficientReasoning [3], which assigns positive rewards to correct responses and zero rewards to incorrect ones, with longer correct responses receiving smaller rewards. A penalty intensity coefficient $\alpha$ is included to fine-tune the compressive intensity. Owing to resource constraints inherent in training reinforcement learning (RL) models, we restrict our evaluation to the Deepseek-R1-Distill-Qwen-1.5B checkpoint [12]. All other experimental configurations are inherited from the EfficientReasoning [3], with rollout budgets specifically set at 10 for AIME24, 3 for MATH500, and 1 for GSM8K. Our intervention is confined to the rollout phase during training. For testing, we rigorously adhere to the original implementation details: environments are constructed using the provided Dockerfile, and inference is conducted with the officially released checkpoints when evaluating EfficientReasoning [3].[1] While the performance on specific datasets showed minor

---

[1]Owing to variations in experimental settings, R1-Distill-Qwen-1.5B exhibits slight performance differences between train-free and train-based experiments. This discrepancy aligns with findings reported in prior studies [16]. We present our results as measured truthfully. Notably, when evaluating the average performance across the three datasets, the outcomes from both experimental environments demonstrate remarkable consistency.

discrepancies, the average performance across the three datasets aligned closely with the results reported in the EfficientReasoning [3].

During the rollout phase, we filter out all "wait" tokens that have confidence scores below 0.3. Reinforcement learning (RL) requires generating samples of varying lengths to strengthen shorter responses. However, indiscriminate intervention across all samples may compromise the learning of negative samples. Negative samples do not need to be shortened in length by suppressing self-affirmation reflections. Therefore, our objective is to suppress self-affirmation reflections in positive samples only. Yet, determining in advance which responses are correct and should be intervened is challenging. Therefore, we adopt an approximate method and intervene with a 25% probability each time.

Table 1: The influence of different thresholds on the original model. The degradation and improvement of performance are marked with Red and Green.

| Models | AIME24 | | AMC23 | | GSM8K | | MATH500 | | Average | Average |
|---|---|---|---|---|---|---|---|---|---|---|
| | Acc↑ | LEN↓ | Acc↑ | LEN↓ | Acc↑ | LEN↓ | Acc↑ | LEN↓ | Acc↑ | LEN↓ |
| R1-Distill-1.5B | 31.7 | 16415.7 | 69.7 | 10370.1 | 75.7 | 676.7 | 82.7 | 5488.0 | 64.9 | 8237.6 |
| Underthink | 27.5 | 16866.6 | 71.3 | 9907.1 | 75.8 | 639.5 | 82.0 | 5641.9 | $64.1_{-0.8\%}$ | $8263.8_{+0.3\%}$ |
| Threshold 0.1 | 32.5 | 15468.9 | 71.9 | 9412.5 | 75.3 | 734.8 | 84.6 | 5268.1 | $66.1_{+1.2\%}$ | $7721.1_{-6.3\%}$ |
| Threshold 0.3 | 27.5 | 16143.3 | 72.5 | 9101.5 | 77.6 | 667.4 | 83.0 | 4893.7 | $65.2_{+0.3\%}$ | $7701.5_{-6.5\%}$ |
| Threshold 0.5 | 27.9 | 14153.8 | 65.6 | 8370.6 | 76.0 | 587.0 | 82.0 | 4924.9 | $62.9_{-2.0\%}$ | $7009.1_{-14.9\%}$ |
| Threshold 0.7 | 26.7 | 14989.6 | 69.7 | 7679.0 | 75.3 | 635.6 | 80.6 | 4549.0 | $63.1_{-1.8\%}$ | $6963.3_{-15.5\%}$ |
| Threshold 0.9 | 28.3 | 14121.2 | 71.6 | 7765.5 | 75.2 | 698.9 | 81.6 | 4191.3 | $64.2_{-0.7\%}$ | $6694.2_{-18.7\%}$ |
| R1-Distill-7B | 50.8 | 13781.8 | 89.7 | 6258.3 | 92.1 | 1533.2 | 92.6 | 4096.8 | 81.3 | 6417.5 |
| Underthink | 53.4 | 12964.6 | 89.7 | 6219.8 | 92.8 | 1405.3 | 93.2 | 4101.4 | $82.3_{+1.0\%}$ | $6172.8_{-3.8\%}$ |
| Threshold 0.1 | 52.1 | 14111.2 | 90.3 | 6205.8 | 91.1 | 1504.4 | 91.4 | 4319.6 | $81.2_{-0.1\%}$ | $6535.3_{+1.8\%}$ |
| Threshold 0.3 | 57.1 | 12852.2 | 89.1 | 6471.4 | 91.8 | 1373.6 | 93.2 | 3760.5 | $82.8_{+1.5\%}$ | $6114.4_{-4.7\%}$ |
| Threshold 0.5 | 50.8 | 12652.2 | 88.4 | 6009.2 | 91.7 | 1397.4 | 92.4 | 3747.2 | $80.8_{-0.5\%}$ | $5951.5_{-7.3\%}$ |
| Threshold 0.7 | 53.3 | 11548.3 | 89.7 | 5426.9 | 92.7 | 1291.3 | 92.4 | 3720.1 | $82.0_{+0.7\%}$ | $5496.6_{-14.3\%}$ |
| Threshold 0.9 | 47.1 | 11477.3 | 87.8 | 5145.2 | 92.2 | 1261.5 | 93.6 | 3365.2 | $80.2_{-1.1\%}$ | $5312.3_{-17.2\%}$ |
| R1-Distill-32B | 70.4 | 11363.4 | 95.0 | 5679.6 | 93.6 | 644.4 | 93.8 | 3640.9 | 88.2 | 5332.1 |
| Underthink | 70.9 | 10744.3 | 95.3 | 5849.0 | 94.0 | 633.0 | 94.8 | 3498.7 | $88.7_{+0.5\%}$ | $5181.3_{-2.8\%}$ |
| Threshold 0.1 | 70.4 | 10955.9 | 96.6 | 5783.8 | 93.6 | 663.4 | 93.6 | 3653.4 | $88.6_{+0.4\%}$ | $5264.2_{-1.3\%}$ |
| Threshold 0.3 | 72.1 | 11170.5 | 95.3 | 5929.7 | 94.0 | 629.4 | 94.4 | 3456.5 | $89.0_{+0.7\%}$ | $5296.5_{-0.6\%}$ |
| Threshold 0.5 | 68.8 | 10350.5 | 94.7 | 5788.8 | 94.1 | 628.4 | 93.6 | 3239.2 | $87.8_{-0.4\%}$ | $5001.7_{-6.2\%}$ |
| Threshold 0.7 | 70.0 | 9747.5 | 96.9 | 5232.1 | 93.5 | 625.5 | 92.6 | 3339.2 | $88.2_{+0.0\%}$ | $4736.1_{-11.1\%}$ |
| Threshold 0.9 | 66.3 | 10193.4 | 94.4 | 4774.4 | 94.2 | 593.4 | 93.8 | 3163.9 | $87.2_{-1.0\%}$ | $4681.3_{-12.2\%}$ |
| QwQ-32B | 78.3 | 13709.0 | 98.8 | 7591.2 | 96.5 | 1646.4 | 95.4 | 4267.6 | 92.2 | 6803.5 |
| Underthink | 77.9 | 13579.9 | 97.5 | 7798.2 | 96.4 | 1540.5 | 96.0 | 4319.5 | $92.0_{-0.2\%}$ | $6809.5_{+0.1\%}$ |
| Threshold 0.1 | 79.6 | 13239.4 | 98.8 | 7683.3 | 96.2 | 1617.6 | 95.2 | 4405.7 | $92.4_{+0.2\%}$ | $6736.5_{-1.0\%}$ |
| Threshold 0.3 | 77.5 | 13296.3 | 99.1 | 7392.8 | 96.4 | 1575.3 | 95.8 | 4241.1 | $92.2_{+0.0\%}$ | $6626.4_{-2.6\%}$ |
| Threshold 0.5 | 77.1 | 13241.8 | 95.6 | 7490.8 | 96.5 | 1546.4 | 95.6 | 4225.9 | $91.2_{-1.0\%}$ | $6626.2_{-2.6\%}$ |
| Threshold 0.7 | 77.1 | 12547.0 | 98.4 | 6693.0 | 96.5 | 1491.3 | 96.2 | 4018.4 | $92.1_{-0.1\%}$ | $6187.4_{-9.1\%}$ |
| Threshold 0.9 | 76.7 | 12613.3 | 97.5 | 6652.5 | 96.4 | 1456.9 | 94.4 | 3883.9 | $91.2_{-1.0\%}$ | $6151.6_{-9.6\%}$ |
| Qwen3-32B | 80.4 | 13369.1 | 96.9 | 7092.2 | 96.3 | 1704.2 | 96.4 | 4570.0 | 92.5 | 6683.9 |
| Underthink | 79.6 | 12579.5 | 97.2 | 7082.3 | 96.1 | 1604.9 | 96.0 | 4667.7 | $92.2_{-0.3\%}$ | $6483.6_{-3.0\%}$ |
| Threshold 0.1 | 80.4 | 13054.4 | 97.8 | 7396.5 | 95.5 | 1723.5 | 95.6 | 4665.9 | $92.3_{-0.2\%}$ | $6710.1_{+0.4\%}$ |
| Threshold 0.3 | 82.1 | 12601.8 | 95.6 | 7131.1 | 96.4 | 1627.6 | 95.2 | 4653.4 | $92.3_{-0.2\%}$ | $6503.5_{-2.7\%}$ |
| Threshold 0.5 | 81.7 | 12571.7 | 97.2 | 6898.0 | 96.2 | 1607.7 | 95.0 | 4535.1 | $92.5_{+0.0\%}$ | $6403.1_{-4.2\%}$ |
| Threshold 0.7 | 81.2 | 12751.7 | 98.4 | 6531.2 | 96.1 | 1601.2 | 95.8 | 4316.8 | $92.9_{+0.4\%}$ | $6300.2_{-5.7\%}$ |
| Threshold 0.9 | 78.8 | 12152.2 | 97.2 | 6510.8 | 96.1 | 1561.5 | 95.6 | 4277.5 | $91.9_{-0.6\%}$ | $6125.5_{-8.4\%}$ |

### 5.2.2 Results

The main results as shown in Table 2. As anticipated, our method achieved substantially shorter average lengths while maintaining comparable performance. It should be noted that text length and accuracy are inherently conflicting objectives. The results from EfficientReasoning [3] demonstrate that the stronger the emphasis on shorter text, the more severe the drop in accuracy. The model pursues shorter positive responses at the same intensity, which is approximately equivalent to using a stronger compression intensity. This explains the slight dip in accuracy. However, as shown in Table 2, the accuracy decrease is minimal and still surpasses the baseline (R1-Distill-Qwen-1.5B). Moreover, at the same text length, our method delivers superior performance.

### 5.2.3 Ablation Study

In this section, we compared the different thresholds for filtering "wait" tokens and the impact of the probability of intervention responses. All ablation experiments were conducted based on $\alpha = 0.05$.

Table 2: The results of EfficientReasoning [3] combines our method. ER is the abbreviation of EfficientReasoning [3]. $*$ denote the results from EfficientReasoning [3] paper. The degradation and improvement of performance are marked with Red and Green.

| Models | AIME24 | | MATH500 | | GSM8K | | Average Acc↑ | Average LEN↓ |
|---|---|---|---|---|---|---|---|---|
| | Acc↑ | LEN↓ | Acc↑ | LEN↓ | Acc↑ | LEN↓ | | |
| R1-Distill-1.5B | 28.7 | 15651.0 | 85.1 | 5274.0 | 75.9 | 709.0 | 63.2 | 7211.3 |
| SFT$*$ | 24.3 | 13805.5 | 77.8 | 3701.2 | 77.6 | 508.2 | $59.9_{-3.3\%}$ | $6004.9_{-16.7\%}$ |
| DPO$*$ | 28.7 | 15145.8 | 83.3 | 4478.6 | 76.3 | 831 | $62.8_{-0.4\%}$ | $6818.4_{-5.4\%}$ |
| ER($\alpha = 0.05$) | 30.3 | 9452.9 | 84.2 | 2648.3 | 85.2 | 776.9 | $66.6_{+3.4\%}$ | $4292.7_{-40.5\%}$ |
| ER($\alpha = 0.1$) | 30.7 | 12071.2 | 82.2 | 2652.0 | 82.1 | 628.0 | $65.0_{+1.8\%}$ | $5117.1_{-29.0\%}$ |
| ER($\alpha = 0.2$) | 29.0 | 10043.7 | 83.3 | 2378.6 | 80.3 | 297.3 | $64.2_{+1.0\%}$ | $4239.9_{-41.2\%}$ |
| ER($\alpha = 0.4$) | 26.3 | 8767.6 | 73.3 | 1869.8 | 68.1 | 138.6 | $55.9_{-7.3\%}$ | $3592.0_{-50.2\%}$ |
| Ours($\alpha = 0.05$) | 27.7 | 7712.6 | 83.6 | 2366.2 | 85.7 | 760.1 | $65.7_{+2.5\%}$ | $3613.0_{-49.9\%}$ |
| Ours($\alpha = 0.1$) | 30.0 | 8312.5 | 81.1 | 2103.8 | 79.8 | 368.3 | $63.6_{+0.4\%}$ | $3594.8_{-50.2\%}$ |
| Ours($\alpha = 0.2$) | 26.3 | 7807.4 | 80.1 | 1798.0 | 80.9 | 341.9 | $62.4_{-0.8\%}$ | $3315.8_{-54.0\%}$ |
| Ours($\alpha = 0.4$) | 29.1 | 7268.0 | 75.9 | 1614.4 | 78.2 | 184.8 | $61.1_{-2.1\%}$ | $3022.4_{-58.0\%}$ |

Table 3: Impact of different thresholds for filtering "wait" tokens. The baseline represent EfficientReasoning [3]($\alpha = 0.05$).

| Models | AIME24 | | MATH500 | | GSM8K | | Average Acc↑ | Average LEN↓ |
|---|---|---|---|---|---|---|---|---|
| | Acc↑ | LEN↓ | Acc↑ | LEN↓ | Acc↑ | LEN↓ | | |
| R1-Distill-1.5B | 28.7 | 15651.0 | 85.1 | 5274.0 | 75.9 | 709.0 | 63.2 | 7211.3 |
| Baseline | 30.3 | 9452.9 | 84.2 | 2648.3 | 85.2 | 776.9 | 66.6 | 4292.7 |
| Threshold-0.1 | 29.7 | 12249.9 | 80.8 | 3031.8 | 81.3 | 538.5 | 63.9 | 5273.4 |
| Threshold-0.3 | 27.3 | 7918.9 | 82.5 | 2269.1 | 85.8 | 703.4 | 65.2 | 3630.5 |
| Threshold-0.5 | 29.0 | 9457.8 | 80.5 | 2241.6 | 82.2 | 478.8 | 63.9 | 4059.4 |
| Threshold-0.7 | 30.3 | 8695.8 | 80.7 | 2358.2 | 83.4 | 632.0 | 64.8 | 3895.3 |
| Threshold-0.9 | 30.3 | 9609.1 | 83.5 | 2883.2 | 82.5 | 676.9 | 65.5 | 4389.8 |

**Impact of different threshold**: In this experiment, the probability of intervention responses is fixed on 50%. As shown in Table 3, when the threshold is below 0.9, the model's performance initially increases but then decreases. This indicates that excessive filtering of self-affirmation reflections can significantly degrade performance. However, when the threshold is set to 0.9, performance improves due to increased length. Overall, there is no linear relationship between the threshold and length. We attribute this to the uncontrollable attribute of the RL training process.

Table 4: Impact of different the probability of intervention responses. The baseline represent EfficientReasoning [3]($\alpha = 0.05$).

| Models | AIME24 | | MATH500 | | GSM8K | | Average Acc↑ | Average LEN↓ |
|---|---|---|---|---|---|---|---|---|
| | Acc↑ | LEN↓ | Acc↑ | LEN↓ | Acc↑ | LEN↓ | | |
| R1-Distill-1.5B | 28.7 | 15651.0 | 85.1 | 5274.0 | 75.9 | 709.0 | 63.2 | 7211.3 |
| Baseline | 30.3 | 9452.9 | 84.2 | 2648.3 | 85.2 | 776.9 | 66.6 | 4292.7 |
| Probability-25% | 27.7 | 7712.6 | 83.6 | 2366.2 | 85.7 | 760.1 | 65.7 | 3613.0 |
| Probability-50% | 27.3 | 7918.9 | 82.5 | 2269.1 | 85.8 | 703.4 | 65.2 | 3630.5 |
| Probability-75% | 29.3 | 12299.4 | 82.4 | 3114.6 | 82.8 | 610.8 | 64.8 | 5341.6 |
| Probability-100% | 24.0 | 8136.7 | 81.5 | 2144.7 | 81.6 | 497.6 | 62.3 | 3593.1 |

**Impact of different probability**: In this experiment, the thresholds for filtering "wait" tokens fixed on 0.3. As shown in Table 4, as the probability increases, performance exhibits a gradual decline. Overly aggressive intervention in the rollout process may generate shorter negative samples, thereby undermining the model's adversarial learning capabilities.

## 6 Conclusion

Our paper delves into the distinctive features of self-affirmation reflections in the distribution of leading words and their correlation with confidence. Leveraging these insights, we introduce a strategy to suppress critical leading words. Experimental results demonstrate that this method effectively reduces output length in both training-free and training-based scenarios while sustaining or even elevating model performance. While we acknowledge certain limitations (see Appendix A.1), we aim for this study to offer a fresh viewpoint on achieving more precise length compression at step-level efficient reasoning and to improve the efficiency of large reasoning models.

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

# A    Technical Appendices and Supplementary Material

## A.1    Limitations

Two core challenges persist. First, methods for systematically identifying candidate tokens for intervention remain underdeveloped. Second, developing principles to balance dependencies among multiple intervened words represents an open research question. Consequently, our intervention framework in this work focuses solely on the most salient specific tokens. Future research will explore strategies to generalize token intervention approaches, addressing both identification scalability and multi-word coordination.

We further recognize that the foundational mechanisms governing the model's reasoning process, as well as the emergence of redundant reflections during generation, are not yet fully understood. These knowledge gaps persist within the large reasoning model community. As the field matures, we anticipate opportunities to integrate insights from complementary work and conduct more in-depth analyses of these phenomena in subsequent studies.

## A.2    Comparison with concurrent works

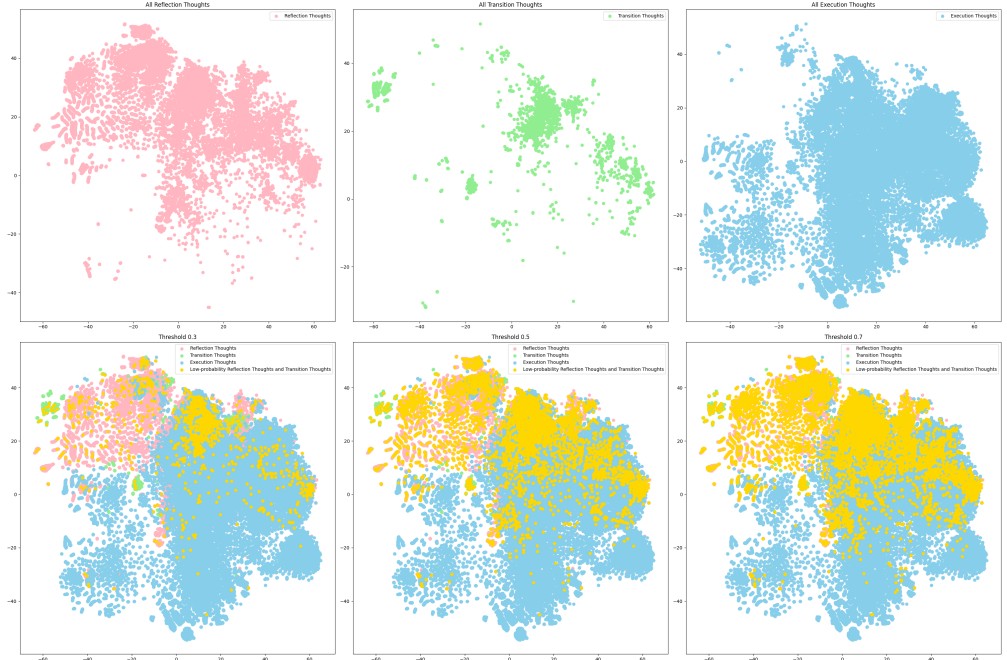

Figure 7: Results of t-SNE visualization of different reasoning thoughts in the SEAL [6]. We additionally visualized "reflection thoughts" and "transition thoughts" with confidence levels below the threshold.

We analyze two concurrent excellent works:

**SEAL [6]**: SEAL focuses on modifying activation values to convert reflective steps (termed "reflection thoughts" and "transition thoughts" in their paper) into executive steps (termed "execution thoughts"). For more details, please refer to the SEAL [6]. In contrast, our work identifies which reflective steps can be preserved and which should be removed. We extend SEAL's t-SNE visualization by highlighting reflective steps with low confidence (see Figure A.2), successfully distinguishing ambiguous reflective steps overlapping with executive steps that typically indicate model confusion.

It is important to note that SEAL only attempts a maximum output length of 10,000 tokens. This limitation arises because SEAL requires intervention on activation values during the "think" phase (between "<think>" and "</think>"), which presents significant implementation challenges, such as integration with vLLM. In contrast, our method achieves a maximum output length of 32,000 tokens, consistent with DeepSeek's testing configuration. This advancement is attributed to our approach's ability to directly integrate with vLLM.

Table 5: Comparison of Underthink [40] and our method. $\alpha$ and $\beta$ represent the intensity and length of the intervention respectively. For more details, please refer to Underthink [40].

| Models | AIME24 | | AMC23 | | GSM8K | | MATH500 | | Average | Average |
| --- | --- | --- | --- | --- | --- | --- | --- | --- | --- | --- |
| | Acc↑ | LEN↓ | Acc↑ | LEN↓ | Acc↑ | LEN↓ | Acc↑ | LEN↓ | Acc↑ | LEN↓ |
| R1-Distill-1.5B | 31.7 | 16415.7 | 69.7 | 10370.1 | 75.7 | 676.7 | 82.7 | 5488.0 | 64.9 | 8237.6 |
| Threshold 0.1 | 32.5 | 15468.9 | 71.9 | 9412.5 | 75.3 | 734.8 | 84.6 | 5268.1 | 66.1 | 7721.1 |
| Threshold 0.3 | 27.5 | 16143.3 | 72.5 | 9101.5 | 77.6 | 667.4 | 83.0 | 4893.7 | 65.2 | 7701.5 |
| Threshold 0.5 | 27.9 | 14153.8 | 65.6 | 8370.6 | 76.0 | 587.0 | 82.0 | 4924.9 | 62.9 | 7009.1 |
| Threshold 0.7 | 26.7 | 14989.6 | 69.7 | 7679.0 | 75.3 | 635.6 | 80.6 | 4549.0 | 63.1 | 6963.3 |
| Threshold 0.9 | 28.3 | 14121.2 | 71.6 | 7765.5 | 75.2 | 698.9 | 81.6 | 4191.3 | 64.2 | 6694.2 |
| $\alpha = 1, \beta = 200$ | 28.8 | 16689.0 | 68.1 | 10315.2 | 76.9 | 729.9 | 84.0 | 5375.0 | 64.4 | 8277.3 |
| $\alpha = 1, \beta = 600$ | 25.4 | 17709.3 | 72.5 | 9776.8 | 75.4 | 736.6 | 86.6 | 5117.2 | 65.0 | 8335.0 |
| $\alpha = 1, \beta = 1000$ | 31.3 | 16736.7 | 70.3 | 10313.4 | 75.8 | 601.9 | 84.8 | 5404.9 | 65.5 | 8264.2 |
| $\alpha = 1, \beta = +\infty$ | 29.6 | 15053.0 | 71.9 | 8445.1 | 75.1 | 615.6 | 83.8 | 4804.9 | 65.1 | 7229.6 |
| $\alpha = 3, \beta = 200$ | 27.9 | 16278.8 | 70.6 | 9100.6 | 76.1 | 748.7 | 84.2 | 5433.1 | 64.7 | 7890.3 |
| $\alpha = 3, \beta = 600$ | 27.5 | 16866.7 | 71.3 | 9907.1 | 75.8 | 639.5 | 82.0 | 5641.9 | 64.1 | 8263.8 |
| $\alpha = 3, \beta = 1000$ | 27.9 | 16497.6 | 68.4 | 10085.1 | 74.3 | 671.0 | 81.8 | 5533.4 | 63.1 | 8196.8 |
| $\alpha = 3, \beta = +\infty$ | 28.8 | 13490.3 | 68.4 | 7167.3 | 75.2 | 602.5 | 83.4 | 4575.3 | 63.9 | 6458.8 |
| $\alpha = 10, \beta = 200$ | 28.8 | 17063.6 | 67.8 | 9989.1 | 75.7 | 692.8 | 83.8 | 5612.0 | 64.0 | 8339.4 |
| $\alpha = 10, \beta = 600$ | 30.4 | 15733.6 | 66.9 | 9917.3 | 75.0 | 635.9 | 82.4 | 5633.3 | 63.7 | 7980.0 |
| $\alpha = 10, \beta = 1000$ | 32.9 | 16708.9 | 70.6 | 9649.2 | 75.5 | 566.9 | 83.2 | 5214.8 | 65.6 | 8035.0 |
| $\alpha = 10, \beta = +\infty$ | 29.2 | 13938.6 | 71.9 | 7276.2 | 75.9 | 648.0 | 82.0 | 4405.8 | 64.7 | 6567.2 |

**Underthink [40]**: This paper aims to address the lack of path exploration capability in models by adjusting the logits of certain reflective leading words within a fixed window. This approach encourages models to explore current reasoning paths more thoroughly during the exploration process. Our focus differs. We aim to suppress self-affirmation reflections in the whole output to achieve more concise reasoning. Whether it is during the exploration process (self-affirmation reflection after already correct steps) or after the completed exploration (self-affirmation reflection after outputting correct answers). Notably, while this work provides a preliminary analysis of underthinking, our work offers a more in-depth and specific analysis of self-affirmation reflections, presenting a fresh perspective on efficient reasoning area.

We also compared our method with Underthink [40], and the results are presented in Table 5 and Table 1. To ensure a fair comparison, we only intervened on the same tokens (all "wait" tokens) as described in the main text. In this context, $\alpha$ and $\beta$ denote the intensity and window length of the intervention, respectively. Underthink [40] recommends parameters of $\alpha = 3$ and $\beta = 600$, which are also the default parameters in Table 1. As shown in Table 5, the phenomenon identified in this paper also contributes to the improvement of Underthink [40]. For instance, when the intervention window length $\beta$ in Underthink [40] is set to $+\infty$, the performance remains nearly unchanged and sometimes even improves, while the outputs become shorter. However, compared with Underthink [40], we consider that intervening based on probability is more straightforward and better aligned with the analysis in our paper. Therefore, we ultimately chose not to directly intervene in the logits.

### A.3 Template

There are the templates we use. During the second judgment phase, the current reflection and the sequence of steps spanning from current reflection step to the subsequent reflection step are processed simultaneously.

**The first judgment template**

"""

Current step: {str1}

Please help me determine the function of the current step.

Is the current step a reflective behavior?

Output the answer directly to <answer></answer>, for example, <answer>Yes</answer> or <answer>No</answer>.

"""

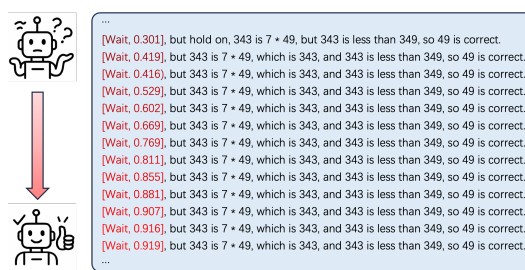

**The second judgment template**

"""

The previous steps: {str1}

The initial step of reflection: {str2}

The subsequent steps of reflection: {str3}

Please help me judge the role of the reflection steps. Is the result of the reflection affirms the previous content? Output your answer directly to <answer></answer>, for example, <answer>Yes</answer> or <answer>No</answer>.

"""

Figure 8: An example where output gets trapped in a loop.

## A.4 Influence of the tail repeated reflections

Figure 8 illustrates a case where the model becomes trapped in a loop. The correct answer is 49, yet the model repeatedly performs self-affirmation reflection and enhances the probability of the leading word. As demonstrated in Figures 9 and 10, the repeated steps filtered to the end of the output significantly impact the statistical results. This presents an intriguing phenomenon worth in-depth investigation. Moreover, the intervention proposed in this paper can partially address this issue. Notably, despite this issue, Self-affirmation reflections still retains significantly confidence bias in leading tokens compared to other reflections. This proves the generalization of our discovery.

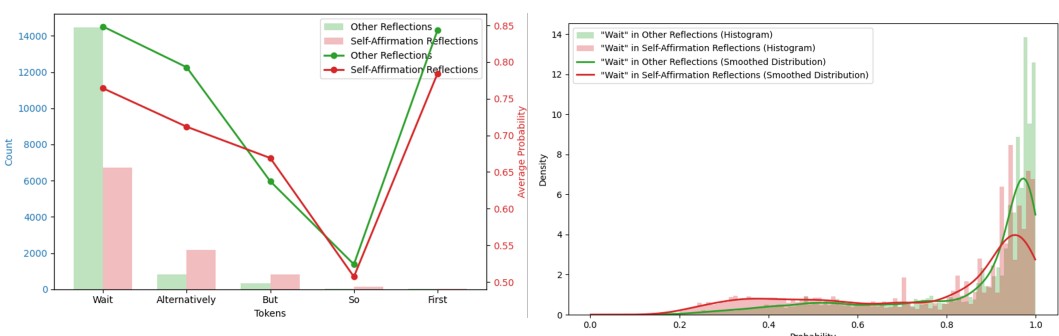

Figure 9: The frequency (via bar chart) and average confidence scores (via line plot) of the first words in all reflective sentences before filtering out the repetitive reflections at the tail.

Figure 10: The distribution of "Wait" in the two types of reflections before filtering out the repetitive reflections at the tail.

## A.5 Discussion on interfered tokens

In this section, we evaluate the impact of intervening on different tokens for R1-Distill-Qwen-1.5B [12]. Results in Table 6 demonstrate consistent patterns across settings. When threshold values are small, all results align with our hypothesis in Section 5.1: removing the Self-Affirmation Reflection does not significantly degrade performance but generates shorter outputs. This behavior is observed in interventions targeting "wait" tokens (Threshold=0.9), "wait" + "alternatively" tokens (Threshold=0.5), and "wait" + "alternatively" + "but" tokens (Threshold=0.5). Increasing threshold values shorten outputs at the cost of reduced performance.

For fixed thresholds, output length decreases progressively as the number of intervened tokens increases. However, addressing dependencies between multiple tokens remains challenging due to combinatorial complexity. Notably, "wait" tokens exhibits statistically significant effects in our analysis. Given these factors, we defer exploration of multi-token interactions to future work and focus primarily on "wait" tokens in this paper.

Table 6: Comparison of different interfered tokens. Baseline represent the results of R1-Distill-Qwen-1.5B [12].

| Models | AIME24 | | AMC23 | | GSM8K | | MATH500 | | Average | Average |
|---|---|---|---|---|---|---|---|---|---|---|
| | Acc↑ | LEN↓ | Acc↑ | LEN↓ | Acc↑ | LEN↓ | Acc↑ | LEN↓ | Acc↑ | LEN↓ |
| Baseline | 31.7 | 16415.7 | 69.7 | 10370.1 | 75.7 | 676.7 | 82.7 | 5488.0 | 64.9 | 8237.6 |
| "wait" tokens | | | | | | | | | | |
| Threshold 0.1 | 32.5 | 15468.9 | 71.9 | 9412.5 | 75.3 | 734.8 | 84.6 | 5268.1 | 66.1 | 7721.1 |
| Threshold 0.3 | 27.5 | 16143.3 | 72.5 | 9101.5 | 77.6 | 667.4 | 83.0 | 4893.7 | 65.2 | 7701.5 |
| Threshold 0.5 | 27.9 | 14153.8 | 65.6 | 8370.6 | 76.0 | 587.0 | 82.0 | 4924.9 | 62.9 | 7009.1 |
| Threshold 0.7 | 26.7 | 14989.6 | 69.7 | 7679.0 | 75.3 | 635.6 | 80.6 | 4549.0 | 63.1 | 6963.3 |
| Threshold 0.9 | 28.3 | 14121.2 | 71.6 | 7765.5 | 75.2 | 698.9 | 81.6 | 4191.3 | 64.2 | 6694.2 |
| "wait" + "alternatively" tokens | | | | | | | | | | |
| Threshold 0.1 | 25.8 | 16756.1 | 66.9 | 10206.1 | 74.6 | 679.0 | 83.4 | 5458.2 | 62.7 | 8274.8 |
| Threshold 0.3 | 30.0 | 16393.9 | 69.4 | 8903.5 | 75.4 | 703.0 | 83.8 | 4944.0 | 64.6 | 7736.1 |
| Threshold 0.5 | 29.2 | 14704.1 | 72.8 | 7606.9 | 76.0 | 567.2 | 84.0 | 4443.0 | 65.5 | 6830.3 |
| Threshold 0.7 | 29.2 | 13302.3 | 69.7 | 6983.7 | 75.8 | 595.9 | 81.8 | 4012.1 | 64.1 | 6223.5 |
| Threshold 0.9 | 26.7 | 12735.7 | 70.0 | 6708.7 | 76.0 | 595.7 | 82.4 | 4086.4 | 63.8 | 6031.6 |
| "wait" + "alternatively" + "but" tokens | | | | | | | | | | |
| Threshold 0.1 | 28.3 | 16280.2 | 71.9 | 9770.4 | 77.1 | 652.8 | 81.8 | 5427.4 | 64.8 | 8032.7 |
| Threshold 0.3 | 30.4 | 14283.2 | 69.4 | 7973.4 | 75.1 | 619.4 | 82.8 | 4546.2 | 64.4 | 6855.5 |
| Threshold 0.5 | 26.7 | 12117.4 | 72.8 | 6287.5 | 76.1 | 528.8 | 82.8 | 3672.2 | 64.6 | 5651.5 |
| Threshold 0.7 | 27.1 | 9525.1 | 68.1 | 5459.7 | 76.1 | 518.9 | 80.4 | 3357.7 | 62.9 | 4715.4 |
| Threshold 0.9 | 24.6 | 9432.5 | 68.4 | 5259.5 | 76.0 | 509.4 | 80.6 | 3221.1 | 62.4 | 4605.6 |

Table 7: The influence of different thresholds on the original model in out-of-domain dataset.

| | R1-Distill-1.5B | | R1-Distill-7B | | R1-Distill-32B | | QwQ-32B | | Qwen3-32B | |
|---|---|---|---|---|---|---|---|---|---|---|
| | Acc↑ | LEN↓ | Acc↑ | LEN↓ | Acc↑ | LEN↓ | Acc↑ | LEN↓ | Acc↑ | LEN↓ |
| Baseline | 33.8 | 10328.9 | 50.9 | 9264.1 | 60.0 | 6723.2 | 62.7 | 9103.7 | 67.7 | 8086.5 |
| Threshold 0.1 | 35.6 | 9288.1 | 48.3 | 8927.4 | 60.9 | 6761.1 | 64.9 | 8934.5 | 68.0 | 8270.6 |
| Threshold 0.3 | 36.5 | 9411.5 | 50.7 | 8462.5 | 60.6 | 6539.8 | 64.9 | 8610.2 | 68.7 | 7973.7 |
| Threshold 0.5 | 33.7 | 9083.3 | 47.1 | 8055.1 | 60.5 | 6123.7 | 63.1 | 8355.0 | 68.8 | 7826.6 |
| Threshold 0.7 | 34.9 | 8896.1 | 46.5 | 8637.6 | 60.6 | 6245.8 | 63.6 | 7987.2 | 67.4 | 7771.4 |
| Threshold 0.9 | 32.6 | 10850.9 | 51.0 | 9498.3 | 58.9 | 6953.1 | 63.4 | 7597.0 | 68.0 | 7571.6 |

## A.6 Results of out-of-domain dataset

We additionally evaluate our model on out-of-domain test data using the GPQA-Diamond benchmark [30]. GPQA-Diamond serves as a rigorous evaluation dataset designed to assess models' capacity for deep reasoning and domain expertise. This dataset represents the highest-quality resource within the GPQA series, comprising 198 graduate-level or competition-level multiple-choice questions. The questions primarily focus on core STEM disciplines including biology, physics, and chemistry, presenting complex problems that require sophisticated reasoning abilities. We sampled four responses for each question and restricted the output length to 32k tokens. We report the average accuracy (Acc) and average token count (LEN) per response. As shown in Table 7, our method generates significantly shorter outputs on out-of-domain dataset while maintaining competitive performance.

## A.7 Implementation in vLLM [19]

```python
from transformers import AutoTokenizer, AutoModelForCausalLM
from vllm import LLM, SamplingParams
import torch
from functools import partial
import os
os.environ["VLLM_USE_V1"] = "0" # we use latest vLLM version for
                                # testing Qwen3. In order to use
                                # logits_processors, we need to set
                                # this to 0

def prob_adjustment(token_ids, logits, adjust_ids, values, threshold):
    assert len(logits.shape) == 1
    probs = torch.softmax(logits, dim=-1)
```

```
586        logits[adjust_ids.to(logits.device)] = torch.where(probs[
587                                          adjust_ids.to(logits.device)] <
588                                           threshold, values, logits[
589                                          adjust_ids.to(logits.device)])
590        return logits
591
592  # load the tokenizer and the model
593  deepseek_1_5b_path="your path"
594  tokenizer = AutoTokenizer.from_pretrained(deepseek_1_5b_path)
595  llm = LLM(model=deepseek_1_5b_path, tensor_parallel_size=4)
596
597  # prepare the model input
598  Question = "Carlos went to a sports store to buy running shoes.
599                                          Running shoes were on sale, with
600                                          prices reduced by $20\%$ on every
601                                          pair of shoes. Carlos also knew
602                                          that he had to pay a $7.5\%$ sales
603                                          tax on the discounted price. He had
604                                           $$43$ dollars. What is the
605                                          original (before discount) price of
606                                           the most expensive shoes he could
607                                          afford to buy?"
608  Template = '<|begin_of_sentence|><|User|>Please reason step by step,
609                                          and put your final answer within \\
610                                          boxed{{}}. Question: {input}<|
611                                          Assistant|><think>\n'  # from
612                                          EfficientReasoning
613  Input_text = Template.format(input=Question)
614
615  # prepare the sampling params, just for a example without any
616                                          randomness. During the formal test,
617                                           we strictly adhered to the
618                                          official configuration of Deepseek.
619  ori_sampling_params = SamplingParams(
620      max_tokens=32768,
621      n=1,
622  )
623  our_sampling_params = SamplingParams(
624      max_tokens=32768,
625      n=1,
626      logits_processors=[
627          partial(
628              prob_adjustment,
629              adjust_ids=torch.tensor([11489,  3783, 14190, 13824]),
630              values=float('-inf'),
631              threshold=0.3
632          )
633      ]
634  )
635
636  # test
637  ori_output = llm.generate(Input_text, sampling_params=
638                                     ori_sampling_params)
639  our_output = llm.generate(Input_text, sampling_params=
640                                     our_sampling_params)
641  ori_length = len(tokenizer.encode(ori_output[0].outputs[0].text,
642                                     add_special_tokens=False))
643  our_length = len(tokenizer.encode(our_output[0].outputs[0].text,
644                                     add_special_tokens=False))
645  print("original length:", ori_length)
646  print("our length:", our_length)
647
```

We provide the simplest implementation method in vLLM [19]. However, for a perfect running speed, we suggest achieving it by directly creating a new CustomSampler class. For example,

```
650
651    llm.llm_engine.model_executor.driver_worker.model_runner.model.sampler
652                                    = CustomSampler(...). # when
653                                    tensor_parallel_size = 1
654
```

