# OpenReview forum: "Efficient Reasoning Through Suppression of Self-Affirmation Reflections in Large Reasoning Models"
_NeurIPS.cc/2025/Conference — Submitted to NeurIPS 2025_

### Official Review · Reviewer_cdvw · 2025-07-02

**Clarity:** 3
**Significance:** 1
**Originality:** 3
**Rating:** 3
**Confidence:** 5

**Summary:**

This paper presents a training-free method to shorten CoT through suppressing self-affirmation reflections. The intuition is that self-affirmation reflection is often redundant and does not provide further information. Through analysis, the authors found that the probability of the leading word for those self-affirmation reflections is lower than that of other reflections, thus the paper proposes to zero-out the probability of those leading reflection words then their probability is too low, to suppress self-affirmation behaviors.

**Questions:**

NA

**Ethical Concerns:**

["NO or VERY MINOR ethics concerns only"]

**Final Justification:**

This paper proposes a decoding approach to suppress self-affirmation reflections of LLMs for more efficient reasoning. This paper presents some interesting analysis about this phenomenon. However, my biggest concern is that the approach does not really make reasoning that much more efficient, thus its practicality is very limited. For example, across both training-free and training-based experiments in both Table 1 and Table 2 of this paper, it reduces tokens by around 500 tokens if requiring no accuracy loss, and by 1000 tokens with a noticeable accuracy loss. This is based on models with 6000-8000 tokens of reasoning on average. I am not satisfactory with this empirical result, thus I lean towards the rejection side.

**Limitations:**

yes

**Quality:**

2

**Strengths And Weaknesses:**

### Strengths

1. Redundant CoT is a critical issue for thinking models and needs more efforts to deal with. It is nice that the proposed approach is training free.
2. Experiments are conducted across multiple reasoning models.


### Weaknesses

1. Generally I think self-affirmation is useful and should not be suppressed just by heuristics, as it may help future reasoning steps judge which steps are more likely to be correct. I think this is reflected by a mild performance decrease of the proposed approach in Table 1 – actually, my biggest concern is that the proposed approach is not very effective and only reduces the CoT length by around 10%-18% most of the time. This is only a slight reduction of the highly redundant thinking process and this small reduction comes with a performance decrease.
2. Line 201-204, I don’t think the authors should draw any conclusion just from a qualitative case study.

---

> ### Author Rebuttal · Authors · 2025-07-30
>
> Thank you for your responses.
>
> # For Weakness 1
>
> ## About "Generally I think self-affirmation is useful and should not be suppressed just by heuristics, as it may help future reasoning steps judge which steps are more likely to be correct."
>
> Thank you for your response. We'd like to offer further clarification. **Self-Affirmation Reflections primarily emphasize affirming prior content in reflection steps, but they do not guarantee that the preceding content is absolutely correct.** While we do not rule out the possibility that future reasoning steps may benefit from such reflections, understanding the deeper mechanisms of reflective behaviors, such as the relationships between different steps, remains a well-known challenge in the field. As noted in our Limitations section, this lies beyond the scope of this paper. The core focus of our paper is to highlight a counterintuitive observation (optimized models unexpectedly generate longer responses) and our subsequent analysis from this phenomenon (what kinds of reflective reasoning can be pruned without degrading model performance, and when such pruning is safe). Ultimately, as Reviewer w67Y noted，our work provides the first in-depth analysis of the Self-Affirmation Reflection phenomenon. For additional discussion, we refer you to our response to Reviewer 32QW, Weakness 1.
>
> Second, **we suppress self-affirmation reflections only when their influence on other reflections is minimal (but not entirely removed).** While you attribute the performance drop to the suppression of self-affirmation reflections, we argue that the decline is more likely due to inappropriate thresholding, which leads to excessive suppression of both self-affirmation reflections and other reflections. Therefore, **the performance degradation cannot be solely attributed to the suppression of self-affirmation reflections.** Importantly, **both our analysis and experiments indicate that self-affirmation reflections can be partially suppressed.** Table 1 presents an ablation study. As shown in some results, partially suppressing self-affirmation reflections results in comparable or improved performance with reduced output length, suggesting that partial suppression of self-affirmation reflections yields net benefits.
>
> ## About "10%-18% is only a slight/small reduction comes with a performance decrease"
>
> First, we achieves this reduction in a training-free manner, with negligible computational overhead. Whether a 10% reduction in length constitutes a slight/small improvement remains open to discussion. To validate the effectiveness of our insights, we also conduct training-based experiments. Notably, **our training-based experiments achieves a significantly larger reduction (up to 50.2%) with no accuracy degradation, which are acknowledged by Reviewers 32QW and u9gx.**
>
> We have also included new methods for comparison: CoD (arXiv:2502.18600) and CCot (arXiv:2407.19825) in the training-free setting, and Spirit-SFT and Spirit-SimPO (arXiv:2502.13260) in the training-based setting. For more detailed results, please refer to our response to Reviewer 32QW, Weakness 2. Compared with these baselines, our method achieves more substantial reductions in length while maintaining or even improving performance. These results further demonstrate that conditionally removing self-affirmation reflections can be acceptable.
>
> |Train-free setting|Acc|Length|
> |-|-|-|
> |Deepseel-Distill-1.5B|64.9|8237.6|
> |Ours|65.2(**+0.3%**)|7701.5(**-6.5%**)|
> |CoD|60.5(**-4.4%**)|6912.5(**-16.1%**)|
> |CCot|56.4(**-8.5%**)|8101.9(**-1.6%**)|
> |Deepseel-Distill-32B|88.2|5332.1|
> |Ours|88.2(**-0.0%**)|4736.1(**-11.1%**)|
> |CoD|84.1(**-4.1%**)|3765.7(**-29.3%**)|
> |CCot|87.7(**-1.3%**)|5259.4(**-1.3%**)|
>
> |Train-base setting|Acc|Length|
> |-|-|-|
> |R1-Distill-1.5B|63.2|7211.3|
> |Ours(α=0.05)|65.7(**+2.5%**)|3613.0(**-49.9%**)|
> |Ours(α=0.1)|63.6(**+0.4%**)|3594.8(**-50.1%**)|
> |Spirit-SFT|65.8(**+2.6%**)|6338.3(**-12.1%**)|
> |Spirit-SimPO|63.9(**+0.7%**)|4401.6(**-38.9%**)|
>
> # For Weakness 2
>
> Thank you for your thoughtful suggestion and careful observation.
>
> (1) We acknowledge that our initial manuscript did not provide a detailed analysis of this issue. In the original version, Figures 5 and 6 focused solely on the quantitative results of the differences of probabilities. To complement these results with qualitative insights, we included Figure 4. Through manual inspection, we observed that the two types of reflections differ not only in the probabilities of the leading words but also in their potential behaviors at the positions where these words appear. Therefore, in addition to showing the probability differences of the leading words, Figure 4 further illustrates the top 8 words and the behaviors they represent. We acknowledge that this issue was inadvertently overlooked in our earlier submission and now provide a preliminary quantitative analysis.
>
> (2) Measuring diversity is inherently challenging. A straightforward approach is to predefine a set of action categories and count how many distinct categories are present, which is the approach we used in Figure 4. However, comprehensively defining action categories is non-trivial. As an approximation, we define three coarse-grained categories based on the paper's context: self-affirmation reflections, other reflections (e.g., necessary reflections), and other actions (e.g., continuing to solve the problem). Similar to Figure 4, we first count how many action categories are present among the top-8 predicted tokens at the leading word position. The following table reports diversity (the first number, defined as the average number of distinct action categories, ranging from 1 to 3). The results show that across all threshold ranges, the leading word positions of Self-Affirmation Reflection consistently exhibit higher diversity. This supports our judgment in the main text.
>
> We further analyze the proportion of each action category, focusing on the latter three numbers in brackets, which represent, from left to right, the proportion of self-affirmation reflections, other reflections, and other actions. Interestingly, the leading word position of other reflections rarely includes self-affirmation reflections actions. In contrast, the self-affirmation reflection position is associated with a more diverse set of actions, encompassing self-affirmation reflections, other reflections, and other actions. The latter two are particularly critical for ensuring answer correctness and advancing problem-solving progress. This asymmetry indirectly supports our insight in this paper.
>
> |Top-1 Leading Words Probability Range|0.0–0.3|0.3–0.5|0.5–0.7|0.7–0.9|0.9–1.0|
> |-|-|-|-|-|-|
> |Other Reflection|**2.16** (3.0% / 43.3% / 53.7%)|**2.10** (3.0% / 49.2% / 47.8%)|**1.99** (3.0% / 58.9% / 38.1%)|**1.79** (2.6% / 70.8% / 26.6%)|**1.32** (1.9% / 86.1% / 12.0%)|
> |Self-Affirmation Reflection|**2.87** (23.5% / 39.9% / 36.6%)|**2.84** (23.4% / 41.0% / 35.6%)|**2.74** (25.8% / 41.1% / 33.1%)|**2.56** (31.2% / 42.1% / 26.7%)|**1.72** (65.4% / 23.5% / 11.1%)|
>
> (3) We further analyze the impact of suppressing specific actions. Given the effectiveness of a threshold of 0.3, we apply suppression to the first predicted token under this threshold and observe how the model’s output changes.
>
> For Self-Affirmation Reflection, when it is suppressed, the distribution of three action becomes 10.9% / **47.6%** / **41.5%**, corresponding to self-affirmation reflections, other reflections, and other actions, respectively. This indicates that when Self-Affirmation Reflection is suppressed, it often transitions into Other Reflection or Other Action. The former helps maintain answer correctness, while the latter facilitates task progression and is partially correlated with output length. These results suggest that Self-Affirmation Reflection can be partially suppressed without severely compromising output quality. Similarly, when Other Reflection is suppressed, the behavior distribution shifts to **4.7%** / 56.8% / 38.5%, again corresponding to self-affirmation reflections, other reflections, and other actions. We observe that when Other Reflection is suppressed, it rarely transitions into Self-Affirmation Reflection. These findings further reinforce the core insight of this work.

---

> > ### Comment · Reviewer_cdvw · 2025-08-06
> >
> > I would like to thank the authors for the detailed response. My biggest concern is indeed on the performance, which I seriously feel quite limited. For example, in the results presented in the rebuttal for DS-distilled 1.5B and 32B models respectively, the reported length reduction is around 500-600 tokens out of 5000-8000 total length. Honestly I don't think this is that meaningful practically.
> >
> > The added training-based results are pretty good, and I encourage the authors to incorporate them to revise the paper. However, as the entire paper of the submission version emphasizes "training-free", the training-based results do not quite fit the submission version of the paper and significant revision is required.
> >
> > For weakness 2, as the authors acknowledged, I think this is a writing issue. I appreciate the detailed explanation which should be incorporated into the paper as well. As such, my main concern on the performance persists, and I would like to maintain my rating.

---

> > > ### Author Response · Authors · 2025-08-06
> > >
> > > We are glad to have addressed most of your concerns. The only remaining issue appears to be the limited performance. We believe there may be a misunderstanding, and we provide further clarification below to ensure a clearer understanding of the paper.
> > >
> > > ## About "My biggest concern is indeed on the performance"
> > >
> > > **First, we sincerely appreciate your recognition of our training-based results.** However, we noticed that your comment referred to them as "the added training-based results." We would like to clarify that not all of these training-based results were newly added in the rebuttal. In fact, **ours results were already presented in the original submission.** According to the suggestions of Reviewer 32QW, we merely compared two additional methods. Your suggestion that we incorporate them into a revised version of the paper may indicate that **this section was inadvertently overlooked. We kindly invite you to revisit Section 5.2 for these results.**
> > >
> > > Second, the opening sentence of the Train-Free Experiments section clearly states the primary goal of Table 1: "We hypothesize that removing low-probability self-affirmation reflection will not degrade model performance. To test this hypothesis, we conducted experiments across several prominent reasoning models." In fact, Table 1 is intended to validate the central insight proposed in our work, not to showcase the best possible results. Regardless of whether the results (500–600 tokens can be reduced in every 5000–8000 token generation with almost no performance loss) is significant, we are puzzled by the repeated emphasis on Table 1 in both the original and current comments. **If the concern were about achieving stronger results, the training-based experiments would have been more relevant**, as also noted by Reviewers 32QW and u9gx: "Length savings up to 18.7 % (training-free) and 50.2 % (training-based) are impressive given minimal accuracy change," and "Extensive experiments on multiple datasets and models show consistent effectiveness, supporting the generality of the approach." **We also emphasized this point using boldface in our rebuttal.**
> > >
> > > Finally, **the main contribution of our paper is not merely to advocate for "train-free" method. Rather, the experiments are designed to demonstrate the generalizability of our insight across different domains, models, and settings.** The core focus of our paper is to highlight a counterintuitive observation (optimized models unexpectedly generate longer responses) and our subsequent analysis from this phenomenon (what kinds of reflective reasoning can be pruned without degrading model performance, and when such pruning is safe). Ultimately, as Reviewer w67Y aptly noted, our work provides the first in-depth analysis of the Self-Affirmation Reflection phenomenon.We hope this work will inspire more fine-grained analyses of overthinking.
> > >
> > > ## About "Honestly I don't think this is that meaningful practically."
> > > Our experiments demonstrate that the insights presented in this work, along with the resulting simple and natural intervention method, are practically meaningful. Compared to intervention strategies that require activation steering, our approach is easier to implement and applicable across different models and settings. **Importantly, even if one considers Table 1 to be of limited value, this does not diminish the significance of other parts in this paper (our observation and analysis, the insights we derive, the proposed intervention and the train-based experiments).** In fact, we find it difficult to understand why, despite our extensive empirical validation of the insights we provide, you appears to focus exclusively on the "train-free" results (an aspect that occupies less than a page out of the nine-page main content).
> > >
> > > ## About "the entire paper of the submission version emphasizes 'training-free', the training-based results do not quite fit the submission version of the paper and significant revision is required."
> > > We have reviewed the usage of the keyword "train-free" in the paper. **On pages 1, 2, and 6, every instance of "train-free" appears alongside "train-based". This reflects our intention to use both train-free and train-based settings to validate the effectiveness of the proposed insight across different experimental conditions.** However, we would like to emphasize that in the first six pages, we never position "train-free" as the primary focus. Instead, the paper begins by analyzing a counterintuitive phenomenon, then focus to self-affirmation reflection, and finally arrives at a simple and effective intervention through progressive analysis. This narrative unfolds naturally, and the distinction between train-free and train-based is not foregrounded in these sections. **In the experimental section (pages 7–9), train-free and train-based experiments are presented in nearly equal proportion.** Therefore, we find it difficult to understand "significant revision is required."

---

> ### Author Response · Authors · 2025-08-04
>
> Dear Reviewer cdvw,
>
> We sincerely appreciate the time and effort you have paid on reviewing our submitted manuscript! Your insightful feedback and reviews have been constructive and invaluable to us, and we really hope our responses could address all your concerns. As the discussion period is approaching the end, we warmly welcome any further questions and discussions from you. We would be delighted to provide additional clarification for you!

---

> ### Comment · Reviewer_cdvw · 2025-08-06
>
> Sorry, my bad in that comment. I did read your paper correctly when I wrote the original review, but when I read the rebuttal I got a wrong impression and forgot the training-based experiments in the original submission (sorry it has been a while since I wrote the original review), thanks for reminding me and the further clarification.
>
> Let me put this way, I somehow have the impression of the method is "training-free" because:
>
> The method itself is indeed training-free like a decoding approach, it doesn't have a "training nature", which means like this approach does not have some key designs/parameters to optimize during training. And the training-based experiments in the paper is to combine other efficient reasoning training approaches with the proposed approach, and the method can further improve the other training-based one.
>
> Is this understanding correct?
>
> I do have a further question on the training-based results in the rebuttal. Is the training in the rebuttal also a combination of the EfficientReasoning and the proposed approach? Because if so, the baseline should be read differently, like the real baseline is EfficientReasoning alone.

---

> > ### Author Response · Authors · 2025-08-06
> >
> > Thank you for your prompt response. Your constructive suggestions help improve the clarity of the paper. Overall, your understanding is correct. While our primary goal is to provide new explanations and insights into the counterintuitive phenomena observed in this paper, the resulting intervention method is indeed does not possess a "training nature." Despite its simplicity, the method is applicable across different training settings.
> >
> > The training-based results reported in the paper are derived from a combination of EfficientReasoning and our proposed approach. Since EfficientReasoning involves sampling multiple responses for a given question and rewarding those that are both correct and short. In contrast to focusing primarily on reward function design, we explore an orthogonal direction by improving the sampling process. This leads to a new approach that yields significant improvements over the original reasoning model.
> >
> > Further analysis is provided in Section 5.2.2. As shown in Table 2, when α = 0.1, our method achieves an absolute reduction in output length of 21.2% compared to EfficientReasoning based on the original reasoning model, and a relative reduction of 29.7% based on EfficientReasoning, while both methods improve accuracy over the original reasoning model. Furthermore, when the length reduction is fixed at 50.2%, our method (α = 0.1) yields a 7.7% absolute accuracy improvement and a 13.8% relative gain over EfficientReasoning (α = 0.4). This also highlights the significance of the insight presented in this work.
> >
> > Additionly, the new training-based results (Spirit) mentioned in the rebuttal do not involve a combination method. Spirit is a standalone approach and is difficult to integrate with EfficientReasoning. This is because Spirit relies on post-processing the full generated output by computing the perplexity of each step and selectively deleting or merging less important steps. This multi-step post-processing is computationally expensive and not suitable for the online rollout setting used in EfficientReasoning. In contrast, our method naturally integrates into the EfficientReasoning.

---

> > > ### Comment · Reviewer_cdvw · 2025-08-07
> > >
> > > Thank you for the response and clarifying that Spirit does not involve a combination method. So the "Ours" rows in the training-based table of the rebuttal is the combined results of EfficientReasoning and the proposed method, right?

---

> > > > ### Author Response · Authors · 2025-08-07
> > > >
> > > > Yes, your understanding is correct. The "Ours" rows in the training-based table of the rebuttal reflect the improvements brought by our insight to existing EfficientReasoning method. This demonstrates a natural and effective application of the insight proposed in our work.
> > > >
> > > > Additionally, Spirit requires post-processing after generating the complete reasoning chain. But our method intervenes efficiently during the reasoning process. This makes Spirit and our method suitable for fundamentally different application scenarios. And our method is also difficult to combine with Spirit.

---

> > > > > ### Comment · Reviewer_cdvw · 2025-08-07
> > > > >
> > > > > Thank you for the clarification! I am willing to raise my score. My overall comment on the performance is: 1. In the training-free experimental setting, the token reduction is limited and doesn't seem very practically useful; 2. In the training-based setting when combining with EfficientReasoning (where the baseline should be EfficientReasoning alone rather than the original model when we judge the contribution of this work), the efficiency boost is slightly better -- reading the absolute tokens from table 2, the proposed approach reduces 500-600 tokens compared with EfficientReasoning often with a performance decrease.
> > > > >
> > > > > I do appreciate the additional analysis and discussion of this phenomenon in this paper, which provides some insights for the audience.

---

> ### Author Response · Authors · 2025-08-08
>
> We sincerely appreciate your valuable feedback and your recognition of our analysis and discussion of self-affirmation reflection phenomenon in the paper. **We also sincerely thank you for your willingness to raise the score.  Our discussion has been very constructive, and we would like to further summarize and supplement it here**:
>
> ## About "practical usefulness"
> Our experiments are designed to verify the feasibility of the insights presented in this work and to demonstrate their **broad applicability across different models** (R1-Distill-1.5B/7B/8B/32B, QwQ-32B, Qwen3-32B), **tasks** (AIME24, AMC23, GSM8K, MATH500, GPQA), and **settings (both training-free and training-based)**. We hope that the insight proposed here will inspire further attention to the overthinking problem **from a fine-grained perspective**. In particular, compared to approaches that require complex activation-steering techniques, our method is **more convenient to implement**, further underscoring the practical utility of our insight.
>
> Regarding the view that "reducing 500–600 tokens is only slightly better," opinions may vary. We would like to make the following additions:
>
> 1. When α = 0.1, our method achieves **an absolute output length reduction of 21.2% compared to ER on the original model, and a relative reduction of 29.7% on ER**. Under a fixed length reduction of approximately 50.2%, our method (α = 0.1) yields a **7.7% absolute accuracy improvement and a 13.8% relative improvement over ER (α = 0.4)**. We believe this is a competitive results.
>
> 2. As described in the paper, the amount of token reduction **varies across tasks of different difficulty levels**. Using ER as the baseline, on AIME24, token counts are reduced by approximately **1500–3700 tokens**. In contrast, on GSM8K, the maximum reduction is around **360 tokens**, and in rare cases token counts may even increase slightly (by fewer than 60 tokens). This discrepancy arises because AIME24 offers greater room for compression, whereas GSM8K inherently provides limited potential for token reduction. Following common practice in the community, we report both token reduction and percentage reduction in average length.
>
> ## About "performance decrease"
>
> It is well known in the literature that reducing response length while improving accuracy constitutes an objective that requires careful handling and balancing. ER, for example, balances this trade-off by introducing a reward function design that penalizes long outputs using α. At the same time, ER’s results show that the stronger the emphasis on reducing response length, the more performance tends to degrade. Unlike ER's focus on reward function design, our method orthogonally improves the rollout stage, thereby reinforcing a model preference for brevity. While this pursuit of conciseness can naturally lead to marginal drops in accuracy under fixed α, our method consistently achieves **a better overall trade-off between length and performance**. This is clearly shown in the following results. Although Tokens per 1% Accuracy (Average length / Average Acc) is not a commonly used metric, these results highlight the strength of our method in achieving a more favorable efficiency–accuracy trade-off in Table 2 and underscore the significance of our findings.
>
> | Method (α)      | Tokens per 1% Accuracy |
> | --------------- | ---------------------- |
> | ER (α = 0.05)   | 64.4550                |
> | ER (α = 0.1)    | 78.7246                |
> | ER (α = 0.2)    | 66.0421                |
> | ER (α = 0.4)    | 64.2576                |
> | Ours (α = 0.05) | **54.9924**            |
> | Ours (α = 0.1)  | **56.5220**            |
> | Ours (α = 0.2)  | **53.1378**            |
> | Ours (α = 0.4)  | **49.4664**            |

---

### Official Review · Reviewer_u9gx · 2025-07-03

**Clarity:** 3
**Significance:** 4
**Originality:** 3
**Rating:** 5
**Confidence:** 2

**Summary:**

This paper presents a simple yet effective train-free intervention method that suppresses low-probability leading words (e.g. “wait”) to reduce self-affirmation reflections in reasoning outputs, achieving shorter outputs without degrading performance. Experiments across multiple models and datasets demonstrate consistent improvements.

**Questions:**

1. Since the method involves directly deleting outputs during inference, what is the rate of misclassification that results in removal of necessary reflections? Does this affect the stability and correctness of complex reasoning tasks?

**Ethical Concerns:**

["NO or VERY MINOR ethics concerns only"]

**Final Justification:**

Overall rating: 5 (Keep)
Confidence: 2 (Keep)

Resolution of key points:

1. The authors commit to adding change indicators in Table 3 and Table 4. Considered resolved (camera-ready should implement).

2. The authors provided detailed responses on the transferability of leading words and the threshold, as well as on misclassification and its impact.

Conclusion: Although the generalization of predefined keywords and a fixed threshold remains limited, the method is a simple, deployable baseline with clear contributions and broad impact. Recommend acceptance.

**Limitations:**

yes

**Quality:**

4

**Strengths And Weaknesses:**

Strengths:

1. The paper proposes a simple train-free suppression method that does not require model modifications, providing an interesting and practical extension for efficient reasoning.

2. Extensive experiments on multiple datasets and models show consistent effectiveness, supporting the generality of the approach.

Weaknesses:

1. Table 3 and Table 4 lack clear change indicators.

2. It is unclear whether leading words differ across datasets or LLMs, and whether the confidence threshold separating self-affirmation from other reflections remains consistent in different contexts. This raises questions about the method's generalizability.

---

> ### Author Rebuttal · Authors · 2025-07-30
>
> Thank you for your thoughtful responses.
>
> # For Weakness 1
>
> Thank you for your reminder. We will add change indicators in the revised version. With the addition of these indicators, Table 3 demonstrates consistent improvements in performance and reductions in length compared to the original model. Table 4 shows that excessive intervention in the RL sampling process leads to significant performance degradation.
>
> # For Weakness 2
> ## About "confidence threshold"
> First, we first observe that **a threshold of 0.3 is generally effective across the mathematics domain.** Furthermore, **consistent phenomenon are also observed in the GPQA** results reported in Table 7 of the appendix. Although the optimal threshold varies across different datasets and models, **the effectiveness of the 0.3 threshold provides support for the generalizability of our insight.** Additionally, **we re-analyzed the results on the GPQA dataset and found trends consistent with those shown in Figures 5 (the difference between frequency and average confidence scores) and 6 (distribution differences).** Due to submission constraints, these results will be included in the appendix. Collectively, these findings validate the generalizability of our method, which **lies in distinguishing self-affirmation reflections from other reflections.** Moreover, understanding how to identify appropriate thresholds to further improve performance remains an important direction for future work.
>
>
> ## About "leading words"
> 1. In fact, we believe the fundamental issue stems from the use of predefined leading words and static thresholds, which maybe lead to poor generalization. This is also the core problem we aim to address now. However, given the prevalent practice in the literatures (e.g., arXiv:2504.15895, arXiv:2505.24863) that using "Wait" in a predefined keywords list and manually tuning associated parameters, we recognize that this is a challenging yet crucial problem. **Since this is not the main focus of our work, we follow common practice in the community.** The core focus of our paper is to highlight a counterintuitive observation (optimized models unexpectedly generate longer responses) and our subsequent analysis from this phenomenon (what kinds of reflective reasoning can be pruned without degrading model performance, and when such pruning is safe). Ultimately, as Reviewer w67Y noted, our work provides the first in-depth analysis of the Self-Affirmation Reflection phenomenon. For additional discussion, we refer you to our response to Reviewer 32QW, Weakness 1. Given the above reason, we chose to use predefined keywords serves solely as a tool for deeper analysis.
>
> 2. **Within the scope of this paper,** although our analysis in Figure 5 identifies several potential leading words, **we focus exclusively on the most representative one to eliminate confounding factors** such as interactions among different leading words and the differences of leading words in different fields. For instance, we also suppress additional leading words as a supplementary experiment in Table 6. We find that introducing more leading words does not consistently lead to improved performance or greater length reduction. This may be due to complex interactions among different leading words. Furthermore, we selected the term 'Wait' as our representative leading word, which aligns with academic convention in this domain and the analyse in our paper. Extensive validation through experiments both within this paper and across numerous other works (e.g., arXiv:2506.08343v1, arXiv:2505.24863) demonstrates its efficacy and generalizability. For instance, our subsequent paper (arXiv:2506.08343v1) also demonstrates that suppressing "Wait" words can enhance model efficiency across a wide range of tasks, including Textual Reasoning, Visual Reasoning, and Video Reasoning. It further demonstrating that **"Wait" is a highly impactful and representative term.** However, we firmly recognize the importance of the issue you raised, which is also an ongoing focus of our current research, as noted in the Limitations section.
>
> 3. **Finally, although our work does not address this challenge, it differs significantly from other contemporaneous studies in several important ways. First, our perspective is novel.** The optimized model unexpectedly produces longer responses. This counterintuitive observation highlights a gap between prior analyses of overthinking and its actual effects. This suggests that the overthink problem requires a more fine-grained and differentiated analysis. **Second, our analysis actually introduces a new angle.** In contrast to existing approaches (e.g., arXiv:2501.18585, arXiv:2506.08343v1, arXiv:2505.24863) that also rely on heuristic token blacklists, our method does not focus on how to intervene, but rather on identifying when an intervention is both possible and unlikely to harm performance. This constitutes an orthogonal contribution compared to prior literature. **Finally, we validate our insights through extensive experiments across a wide range of tasks and models.** In the training-free setting, our insight leads to an improvement with minimal accuracy change and almost no additional cost. In the training-based setting, compared to approache (e.g., arXiv:2502.04463) that focus on designing reward functions, our insight naturally introduces an orthogonal benefit: it enhances the quality of positive samples during the sampling process. This demonstrates the novelty, significance and generality of this paper. For more results in rebuttal stage, please refer to our response to Reviewer w67Y, Weakness 1 (About "universally applicable across all models or tasks").
>
> # For Question 1
>
> ## About "the rate of misclassification that results in removal of necessary reflections"
>
> Thank you for your question. Given the effectiveness of a threshold of 0.3, we report two metrics at a commonly used threshold of 0.3, under the setting described in Section 4.1. The first metric, (Filtered other reflections) / (Filtered other reflections + Filtered self-affirmation reflections), corresponds to the misclassification rate. The second metric, (Filtered other reflections) / (Total other reflections), reflects the extent to which necessary reflections are suppressed.
>
> Please refer to the results in Figure 6. For the first metric, we focus on the region in Figure 6 where the x-axis values are less than 0.3. Specifically, we compute the ratio of the green area to the sum of the green and red areas within this region, resulting in a misclassification rate of 25.7%. For the second metric, we compute the proportion of the green area with x-axis values less than 0.3 relative to the total green area, yielding a suppression rate of 4%.
>
> We further evaluated the transition trends of reflections when interventions were applied. Detailed experimental procedures can be found in our response to Reviewer cdvw, Weakness 2. We predefined three categories of actions: self-affirmation reflections, other reflections (e.g., necessary reflections), and other actions (e.g., continuing to solve the problem). **Although 25.7% of reflections are misclassified, this accounts for only 4% of all Other Reflections.** Nonetheless, **the incorrectly suppressed Other Reflections still have the potential to re-emerge in subsequent turns as new Other Reflections, and they rarely transition into Self-Affirmation Reflections.** Conversely, self-affirmation reflections that are suppressed tend to convert into other reflections or other actions, where the former helps maintain answer correctness, and the latter accelerates problem-solving. This indirectly helps preserve the overall presence of other reflections. Futhermore, **our extensive results also prove that the suppression of only 4% of other reflections has a limited impact on overall performances.**
>
> ## About "stability and correctness"
>
> We have made efforts to validate the stability/correctness of the insights presented in this work. To assess the robustness of our findings, we conducted extensive experiments **across different models (R1-Distill-Qwen 1.5B/7B/32B, QwQ-32B, and Qwen3-32B), differect domains (see Tables 1 and 2, and Appendix A.6), and different settings (both train-free and train-based settings).** Additionally, our experiments on R1-Distill-Llama-8B further confirm the stability and correctness of the findings. For detailed results, please refer to our response to Reviewer w67Y, Weakness 1 (About "universally applicable across all models or tasks"). Therefore, we believe that suppressing self-affirmation reflections during inference does not significantly affect the stability or correctness of complex reasoning tasks.

---

> > ### Comment · Reviewer_u9gx · 2025-08-06
> > **Thank you for your response**
> >
> > Thank you for your detailed and thoughtful responses. Your rebuttal has addressed my concerns. I will maintain my original score.

---

> > > ### Author Response · Authors · 2025-08-07
> > >
> > > Thank you for your valuable reply, we appreciate your effort and time, and we are so glad to hear that your concerns have been addressed.

---

> ### Author Response · Authors · 2025-08-04
>
> Dear Reviewer u9gx,
>
> We sincerely appreciate the time and effort you have invested in reviewing our submission! Your insightful feedback and reviews have been invaluable to us, and we really hope our responses have been helpful to you. As the discussion period is approaching the end, we warmly welcome any further questions and discussions from you. We will be honored to provide additional clarification for your further concerns! Thanks again!

---

### Official Review · Reviewer_32QW · 2025-07-03

**Clarity:** 3
**Significance:** 2
**Originality:** 2
**Rating:** 4
**Confidence:** 3

**Summary:**

The paper addresses a specific cause of over-thinking in LRMs: Self-Affirmation Reflections (SARs), i.e., low-information sentences that simply restate or praise already-correct reasoning steps. By analyzing token-level probabilities, the authors discover that SARs start with a small set of “leading words” (notably wait/Wait) whose logits are systematically low. They exploit this bias with a train-free decoding hack: whenever the next-token probability of a SAR leading word falls below a threshold, its probability is zeroed, preventing the reflection from being generated. A second, train-based variant injects the same rule during RL rollouts. Experiments on four math-reasoning benchmarks and five open models show sizeable length savings (18.7% training-free, 50.2% training-based for R1-Distill-1.5B) with negligible or even positive accuracy change, and the intervention plugs straight into vLLM.

**Questions:**

Please refer to the weaknesses section and address all issues raised.

**Ethical Concerns:**

["NO or VERY MINOR ethics concerns only"]

**Final Justification:**

While I still find the definition of “leading words” proposed in the paper somewhat naive and limited in diversity, I appreciate the authors' attitude and significant efforts in addressing all reviewers' comments during the rebuttal. For this reason, I have decided to raise my score.

**Limitations:**

yes

**Quality:**

2

**Strengths And Weaknesses:**

## Strengths

1. The paper brings attention to SARs, a prevalent and interesting behavior in LRM reasoning.

2. Linking SAR occurrence to low-confidence leading words is original and empirically well supported by Figures 4-6.

3. The decoding-time suppression requires no extra parameters, data, or gradient steps and is drop-in compatible with vLLM and RL training loops.

4. Experiments are extensive.  Results span diverse model sizes and four datasets, plus ablations over thresholds and intervention rates. Length savings up to 18.7 % (training-free) and 50.2 % (training-based) are impressive given minimal accuracy change.

## Weaknesses

1. As SAR is a narrowly targeted phenomenon under overthinking, the proposed fix is conceptually a heuristic token blacklist, and marks limited novelty to the broader literature on efficient LRM reasoning through compressing reasoning chain.

2. In addition to the previous weakness, it seems that Underthink is treated as the main (and only) baseline. However, there exist various approaches for efficient LRM reasoning  through reasoning chain compression, such as Constrained-CoT [1], Chain-of-Draft [2], and LightThinker [3], to name just a few. While not explicitly targeted at reducing SARs, they can implicitly compress those as part of the general reasoning chain.

3. SAR labelling depends on a proprietary 72 B model and is only ~80% accurate (Section 4.1, Line 184), raising doubt about analysis fidelity.

4. Suppressing low-probability wait tokens may also remove beneficial reflections or leave SARs starting with other tokens. The lack of human evaluation leaves answer readability and informativeness unverified.

5. The paper optimises token count but omits runtime, latency, or GPU-cost metrics, which are arguably the true targets of “efficient reasoning”.



[1] Concise Thoughts: Impact of Output Length on LLM Reasoning and Cost. arXiv:2407.19825

[2] Chain of draft: Thinking faster by writing less. arXiv:2502.18600

[3] LightThinker: Thinking Step-by-Step Compression. arXiv:2502.15589

---

> ### Author Rebuttal · Authors · 2025-07-30
>
> Thank you for your valuable insights and suggestions.
> # For Weakness 1
> ## About "a narrowly targeted phenomenon under overthinking"
> We would like to clarify that overthinking is a broad problem. Many contemporary works (e.g., arXiv:2501.12570, arXiv:2502.04463) have addressed this issue by demonstrating dataset-level reductions in output length. These approaches often involve sampling multiple responses for a given question and rewarding those that are both correct and concise, which effectively optimizes the model's output preference. **However, our observation (Optimized models sometimes generate longer responses for certain questions) reveals a counterintuitive phenomenon and suggests a gap between the prior analysis of overthinking and its practical effects**. We argue that addressing overthinking requires a more fine-grained and nuanced analysis, particularly in identifying: **what kinds of reflective reasoning can be pruned without degrading model performance, and when such pruning is safe**. Among the various forms of reflection, we identify SAR as a candidate that can be pruned. When the leading word probabilities of SAR are low, its influence on other reflective components is minimal, making it a safe target for pruning. Therefore, although overthinking has only been loosely defined, **our perspective is not based on existing branches or merely a narrow subproblem of overthinking. Instead, we offer a novel and necessary viewpoint that provides deeper insights into its structure and implications.** As noted by Reviewer w67Y, we "provide the first in-depth analysis of the Self-Affirmation Reflection phenomenon, offering valuable insights for improving reasoning models." We hope this work will inspire more fine-grained analyses of overthinking.
> ## About "a heuristic token blacklist" and "limited novelty"
>
> 1. Because using predefined keywords is the common practice in the vast majority of current works (e.g., arXiv:2504.15895, arXiv:2505.24863, arXiv:2501.18585) in this field. **Our heuristic token blacklist serves solely as a tool for deeper analysis. In contrast to prior studies that focus on how to intervene based on such heuristic token blacklist, our work focuses more on when interventions should occur, with an emphasis on preserving model performance.** In fact, although our analysis in Figure 5 identifies several potential leading words, we focus exclusively on the most representative one to eliminate confounding factors such as interactions among different leading words. We selected the term 'Wait' aligns with academic convention in this domain and the analyse in our paper. Extensive validation through experiments both within this paper and across numerous other works (e.g., arXiv:2506.08343v1) demonstrates its efficacy and generalizability. For a detailed explanation and more results, please refer to our response to Reviewer w67Y, Weakness 1.
>
> 2. **Importantly, our work differs significantly from prior studies that rely on predefined leading words. First, our perspective is novel.** The optimized model unexpectedly produces longer responses. This counterintuitive observation highlights a gap between prior analyses of overthinking and its actual effects. This suggests that the overthink problem requires a more fine-grained and differentiated analysis. **Second, our analysis actually introduces a new angle.** In contrast to existing approaches (e.g., arXiv:2501.18585, arXiv:2506.08343v1, arXiv:2505.24863) that also rely on heuristic token blacklists, our method does not focus on how to intervene, but rather on identifying when an intervention is both possible and unlikely to harm performance. This constitutes an orthogonal contribution compared to prior literature. **Finally, we validate our insights through extensive experiments across a wide range of tasks and models.** In the training-free setting, our insight leads to an improvement with minimal accuracy change and almost no additional cost. In the training-based setting, compared to approache (e.g., arXiv:2502.04463) that focus on designing reward functions, our insight naturally introduces an orthogonal benefit: it enhances the quality of positive samples during the sampling process. This demonstrates the novelty, significance and generality of this paper.
> # For Weakness 2
> Thank you for the suggestion. **In the train-free setting, we additionally compared with CoD (arXiv:2502.18600) and CCot (arXiv:2407.19825).** We did not include LightThinker (arXiv:2502.15589) in our comparison because its current implementation is not well integrated with efficient inference frameworks such as vLLM or SGLang, resulting in slower efficiency under standard reasoning task evaluation settings. We select two models with significantly different model sizes. Due to the limit on the number of submitted characters, we present average results on 4 datasets here. **Compared to other methods, our approach achieves length reduction while maintaining or even improving performance.**
>
> ||Acc|Length|
> |-|-|-|
> |Deepseel-Distill-1.5B|64.9|8237.6|
> |Ours|65.2(**+0.3%**)|7701.5(**-6.5%**)|
> |CoD|60.5(**-4.4%**)|6912.5(**-16.1%**)|
> |CCot|56.4(**-8.5%**)|8101.9(**-1.6%**)|
> |Deepseel-Distill-32B|88.2|5332.1|
> |Ours|88.2(**+0.0%**)|4736.1(**-11.1%**)|
> |CoD|84.1(**-4.1%**)|3765.7(**-29.3%**)|
> |CCot|87.7(**-1.3%**)|5259.4(**-1.3%**)|
>
> **In the train-based setting, we additionally compare our approach with Spirit (arXiv:2502.13260).** Spirit quantifies the importance of each reasoning step via perplexity, removing or merging unimportant steps while preserving the overall coherence of the reasoning process. This shares a similar approach with our method, which removes SAR steps to achieve reasoning compression. On R1-Distill-1.5B, using two standard learning paradigms (SFT and SimPO), we obtain the results shown below. **Our method achieves a significantly higher degree of length compression compared to Spirit.**
>
> ||Acc|Length|
> |-|-|-|
> |R1-Distill-1.5B|63.2|7211.3|
> |Ours(α=0.05)|65.7(**+2.5%**)|3613.0(**-49.9%**)|
> |Ours(α=0.1)|63.6(**+0.4%**)|3594.8(**-50.1%**)|
> |Spirit-SFT|65.8(**+2.6%**)|6338.3(**-12.1%**)|
> |Spirit-SimPO|63.9(**+0.7%**)|4401.6(**-38.9%**)|
> # For Weakness 3
> The reliability of LLM-as-a-judge remains a broadly discussed issue (arXiv:2506.22316v1). Although evaluation criteria vary across tasks, several papers that do report accuracy metrics (arXiv:2306.05685, arXiv:2502.06193, arXiv:2409.11860) suggest that **accuracy or Pearson correlation in the range of 75% to 85% is generally acceptable.** Given the associated cost, we argue that an LLM with approximately ~80% accuracy provides a reliable approximation for our analysis.
>
> We also acknowledge that the validity of our analysis may be affected by the inherent limitations in LLM judgment accuracy. However, we believe that this influence is bounded within a tolerable range. The conclusions derived from this analysis serve as the basis for both the train-free and train-based experiments. **The consistency of the experimental results, in turn, provides indirect support for the fidelity of our analysis.**
>
> **Additionally, on a manually annotated validation set, we observe trends similar to those reported in Figures 5 (the difference between frequency and average confidence scores) and Figure 6 (the difference of distribution).** Taking "Wait" as an example, the average confidence across all other reflections is 0.705, whereas under Self-Affirmation Reflections it drops to 0.497. Due to image upload limitations, we will include new figures in the appendix.
> # For Weakness 4
>
> We acknowledge your assessment in our paper. Here, we further provide a more detailed analysis of the intervention methods.
> 1. We predefine three types of actions: self-affirmation reflections, other reflections (e.g., necessary reflections), and other actions (e.g., continuing to solve the problem). We suppress the first token to observe how the model’s output changes. The detailed experimental procedure can be found in our response to Reviewer cdvw, Weakness 2.
> 2. We also calculate the proportion of other reflections that were incorrectly intervened upon among all other reflections, which is only 4%. The detailed experimental procedure is provided in our response to Reviewer u9gx, Question 1.
>
> These quantitative results show that **mistakenly suppressed Other Reflections (although they account for only 4% of all Other Reflections) still retain some probability of generating another Other Reflections**, thereby offering a degree of compensation. Moreover, **while it is possible for Self-Affirmation Reflections to begin with different tokens, deleted Self-Affirmation Reflections are more likely to shift into Other Reflections or Other Actions.** The former contributes to maintaining answer correctness, while the latter supports task progression and is partially correlated with output length. Together with the experimental findings presented in the paper, these results further demonstrate the effectiveness of our approach.
>
> (Readability and Informativeness) Furthermore, both our human evaluation on 20 samples and findings from prior work (e.g., arXiv:2505.24863, arXiv:2506.08343v1) indicate that the removal of reflections does not noticeably degrade the readability and informativeness of the answers. All of the above findings will be incorporated into revised versions.
> # For Weakness 5
> Our method can be implemented with minimal overhead, requiring only a modification of the probability distribution prior to sampling. We believe this change introduces no significant GPU cost or latency. For instance, we evaluated our approach using Deepseek-Distill-7B on the MATH500 dataset with a maximum output length of 32k tokens, running on a single L20 GPU. In practice, we found the additional latency to be entirely acceptable.
> ||Tokens per second|
> |-|-|
> |Original Speed|989.98|
> |Ours|986.15|

---

> > ### Comment · Reviewer_32QW · 2025-08-07
> >
> > I thank the authors for the detailed responses. For this reason, I will increase my rating to 4.
> >
> > The authors are expected to incorporate all responses into the revised version to strengthen the paper's contributions.

---

> > > ### Author Response · Authors · 2025-08-07
> > >
> > > We greatly appreciate your constructive comments. The additional content supplemented during the rebuttal period will be integrated into the revised version. Thank you very much for your positive feedback and for increasing your score.

---

> ### Author Response · Authors · 2025-08-04
>
> Dear Reviewer 32QW,
>
> We sincerely appreciate the time and effort you have invested in reviewing our submitted manuscript! Your insightful feedback and reviews have been invaluable to us, and we really hope our responses have been helpful to you. As the discussion period is reaching the end, we kindly welcome any further questions and discussions from you. We would be pleased to provide additional clarification!

---

### Official Review · Reviewer_w67Y · 2025-07-12

**Clarity:** 3
**Significance:** 2
**Originality:** 3
**Rating:** 4
**Confidence:** 3

**Summary:**

The paper addresses the challenges faced by large language models in complex reasoning tasks, particularly focusing on the issue of redundant reflection, termed Self-Affirmation Reflection. Despite advancements in reasoning models through methods like reinforcement learning and supervised fine-tuning, these models often generate unnecessarily lengthy reasoning steps, leading to inefficiencies.

**Questions:**

none

**Ethical Concerns:**

["NO or VERY MINOR ethics concerns only"]

**Limitations:**

See above

**Quality:**

3

**Strengths And Weaknesses:**

Strengths:
1. The paper provides the first in-depth analysis of the Self-Affirmation Reflection phenomenon, offering valuable insights for improving reasoning models.

Weaknesses:
1. The proposed method relies heavily on the identification and suppression of leading words, which may not be universally applicable across all models or tasks;
2. The simplicity of the intervention strategy, while advantageous, might overlook more complex underlying factors contributing to reflection behaviors in reasoning models.

---

> ### Author Rebuttal · Authors · 2025-07-30
>
> Thank you for your thoughtful responses.
>
> # For Weakness 1:
>
> ## About "relies heavily on the identification and suppression of leading words"
>
> 1. In general, we acknowledge that relying on predefined leading words lacks universality in theory, as also emphasized in our limitations section. However, using predefined keywords is the common practice in the vast majority of current works (e.g., arXiv:2504.15895, arXiv:2505.24863, arXiv:2501.18585) in this field. This indicates that the problem is challenging. **Since this issue is not the primary focus of our work, we follow the common practice in the community.** The core focus of our paper is to highlight a counterintuitive observation (optimized models unexpectedly generate longer responses) and our subsequent analysis from this phenomenon (what kinds of reflective reasoning can be pruned without degrading model performance, and when such pruning is safe). Ultimately, our work provides the first in-depth analysis of the Self-Affirmation Reflection phenomenon. For a detailed explanation of our focus, please refer to our response to Reviewer 32QW, Weakness 1
>
> 2. **Given the above reason, within the scope of this paper, although our analysis in Figure 5 identifies several potential leading words, we focus exclusively on the most representative one to eliminate confounding factors** such as interactions among different leading words and the differences of leading words in different fields. For instance, we also suppress additional leading words as a supplementary experiment in Table 6. We find that introducing more leading words does not consistently lead to improved performance or greater length reduction. This may be due to complex interactions among different leading words. Therefore, we focus on the most salient token, which serves as a reasonable basis for subsequent analysis.
>
> 3. In fact, we selected the term 'Wait' as our representative leading word, which aligns with academic convention in this domain and the analyse in our paper. Extensive validation through experiments both within this paper and across numerous other works (e.g., arXiv:2506.08343v1, arXiv:2505.24863) demonstrates its efficacy and generalizability. For instance, our subsequent paper (arXiv:2506.08343v1) also demonstrates that suppressing "Wait" words can enhance model efficiency across a wide range of tasks, including Textual Reasoning, Visual Reasoning, and Video Reasoning. It further demonstrating that **"Wait" is a highly impactful and representative term**. In future work, we will further explore automatic methods for identifying and intervening on leading words.
>
> ## About "universally applicable across all models or tasks"
>
> 1. We have made effort to validate the generality of the insights presented in this work. **To assess the robustness of our findings, we conducted extensive experiments across different models (R1-Distill-Qwen 1.5B/7B/32B, QwQ-32B, and Qwen3-32B), differect domains (see Tables 1 and 2, and Appendix A.6), and different settings (both train-free and train-based settings)**.
>
> 2. Apart from the mathematics domain, GPQA is also a commonly used dataset for complex reasoning. We re-analyzed the results on the GPQA dataset and observed trends consistent with those reported in Figure 5 (the difference between frequency and average confidence scores) and Figure 6 (the difference of distribution). Due to submission limitations, we plan to include these results in a future version of the appendix. This suggests that **similar phenomena also occur across different domains.**
>
> 3. **Beyond the Qwen model family, we also tested generalizability across different model architectures by evaluating the LLaMA-series model.** The results are summarized below.
> | |Average Acc on 4 datasets|Average Length|
> |-|-|-|
> |R1-Distill-Llama-8B|75.6|6965.2|
> |Threshold-0.3|76.2(**+0.6%**)|6389.4(**-8.3%**)|
>
> 4. We additionally attempted to reduce self-affirmation reflections on R1-Distill-Qwen-32B by directly modifying the prompt to include: "Only output necessary reflections. Do not output reflections that contain redundant content or merely affirm previous text." The results are summarized below. **The identification and suppression of leading words more accurately reduces output length without compromising performance.**
> | | Average Acc on 4 datasets | Average Length |
> |-|-|-|
> |Baseline|88.2|5332.1|
> |Leading-words-suppress|88.2(**-0.0%**)|4736.1(**-11.1%**)|
> |Prompt-suppress|86.3(**-1.9%**)|4842.8(**-9.2%**)|
>
>
> # For Weakness 2:
>
> We appreciate your thoughtful comment. Our "observation → hypothesis → validation" analysis method is inspired by some impactful works (e.g., arXiv:2412.21187). Following this widely adopted paradigm in the field, our empirical analysis focuses on the probability differences between self-affirmation reflections and other reflections, which naturally motivates the intervention method proposed in this paper. While the experimental results demonstrate its effectiveness, we acknowledge that more complex underlying factors may be involved. Therefore, we deliberately chose to concentrate on the most prominent leading words in our experiments and analysis, rather than including a broader set. This approach helps ensure that the experiment and analysis do not explicitly introduce additional confounding factors, aligning with your perspective.
>
> Furthermore, we provide a more detailed analysis of the intervention methods.
> 1. We predefine three types of actions: self-affirmation reflections, other reflections (e.g., necessary reflections), and other actions (e.g., continuing to solve the problem). We suppress the first token to observe how the model’s output changes. The detailed experimental procedure can be found in our response to Reviewer cdvw, Weakness 2.
> 2. We also calculate the proportion of other reflections that were incorrectly intervened upon among all other reflections, which is only 4%. The detailed experimental procedure is provided in our response to Reviewer u9gx, Question 1.
>
> These quantitative results demonstrate that our intervention effectively reduces the occurrence of Self-Affirmation Reflections, as they are more likely to shift to other reflections and other actions. The former helps maintain answer correctness, while the latter facilitates task progression and is partially correlated with output length. Meanwhile, mistakenly suppressed Other Reflections (which account for only 4% of all Other Reflections) still retain a certain probability of generating another Other Reflection, providing a degree of compensation. Collectively, these findings offer more nuanced support for the effectiveness of our intervention.
>
>
> Finally, the in-depth mechanistic analysis of reflective behavior remains an important yet unresolved problem. In future work, we plan to further investigate the interactions between different leading words and the potential factors they introduce that may trigger reflective behavior.

---

> ### Author Response · Authors · 2025-08-04
>
> Dear Reviewer w67Y,
>
> We sincerely appreciate the time and effort you have invested in reviewing our submission! Your insightful feedback and reviews have been invaluable to us, and we really hope our responses have been helpful to you. As the discussion period is approaching the end, we warmly welcome any further questions and discussions from you. We would be delighted to provide additional clarification!

---

### Decision · Program_Chairs · 2025-09-17

**Decision:**

Reject

**Comment:**

This paper focuses on the overthink phenomenon in reasoning models and presents an analysis of the self-affirmation reflection phenomenon (i.e., redundant affirmation of already-correct prior reasoning steps), which can be pruned to reduce thinking length without degrading accuracy. The paper identifies that when "wait" serves as a low-probability leading token, self-affirmation reflection is more likely to occur compared to other reflection patterns. Based on this observation, the authors propose to mitigate this phenomenon by suppressing low-probability "wait" tokens.

While the analysis of the self-affirmation reflection phenomenon is interesting, reviewers raised several concerns that were not adequately addressed during the discussion period:

1. The use of low-probability "wait" tokens as indicators of self-affirmation reflection lacks convincing generalizability across different models and domains.

   a. In the training-free setting, when the threshold is set low (≤0.3), length reduction is marginal. Only when the threshold exceeds 0.7—which appears to also suppress high-probability "wait" tokens—does length reduction increase to 5-18%. It remains unclear what exactly is being suppressed and whether suppressing self-affirmation reflections is indeed the primary driver of length reduction. As reviewer w67Y noted, this intervention strategy may overlook more complex underlying factors.

   b. Experimental results show inconsistency across datasets. With the threshold set to 0.9, R1-distilled models achieved greater length reduction than at lower thresholds on math datasets (Table 1), yet generated more tokens than the baseline without intervention on GPQA (Table 7).

2. The proposed method demonstrates limited effectiveness in reducing reasoning length. In the training-free setting, it reduces only 500-1000 tokens from models generating 6000-8000 reasoning tokens on average. In the training-based setting, when combined with EfficientReasoning, the latter reduces 3000 tokens from models averaging 7000 reasoning tokens, while the proposed method contributes an additional reduction of merely 500 tokens. As demonstrated in Tables 2 and 3, varying the length penalty in EfficientReasoning results in length changes on the order of hundreds of tokens, while adjusting the threshold in the proposed method yields similarly changes of hundreds of tokens. It's not convinced that the proposed method provides significant complementary improvements.

The recommendation is to reject this paper.